# Learning Representation from Neural Fisher Kernel with Low-rank Approximation

**Ruixiang Zhang**
Mila, Université de Montréal
ruixiang.zhang@umontreal.ca

**Shuangfei Zhai, Etai Littwin, Josh Susskind**
Apple Inc.
{szhai,elittwin,jsusskind}@apple.com

## Abstract

In this paper, we study the representation of neural networks from the view of kernels. We first define the Neural Fisher Kernel (NFK), which is the Fisher Kernel (Jaakkola and Haussler, 1998) applied to neural networks. We show that NFK can be computed for both supervised and unsupervised learning models, which can serve as a unified tool for representation extraction. Furthermore, we show that practical NFKs exhibit low-rank structures. We then propose an efficient algorithm that computes a low rank approximation of NFK, which scales to large datasets and networks. We show that the low-rank approximation of NFKs derived from unsupervised generative models and supervised learning models gives rise to high-quality compact representations of data, achieving competitive results on a variety of machine learning tasks.

## 1 Introduction

Modern deep learning systems rely on finding good representations of data. For supervised learning models with feed forward neural networks, representations can naturally be equated with the activations of each layer. Empirically, the community has developed a set of effective heuristics for representation extraction given a trained network. For example, ResNets (He et al., 2016) trained on Imagenet classification yield intermediate layer representations that can benefit downstream tasks such as object detection and semantic segmentation. The logits layer of a trained neural network also captures rich correlations across classes which can be distilled to a weaker model (Knowledge Distillation) (Hinton et al., 2015).

Despite empirical prevalence of using intermediate layer activations as data representation, it is far from being the optimal approach to representation extraction. For *supervised learning* models, it remains a manual procedure that relies on trial and error to select the optimal layer from a pre-trained model to facilitate transfer learning. Similar observations also apply to *unsupervised learning* models including GANs (Goodfellow et al., 2014), VAEs (Kingma and Welling, 2014), as evident from recent studies (Chen et al., 2020a) that the quality of representation in generative models heavily depends on the choice of layer from which we extract activations as features. Furthermore, although that GANs and VAEs are known to be able to generate high-quality samples from the data distribution, there is no strong evidence that they encode explicit layerwise representations to similar quality as in supervised learning models, which implies that there does not exist a natural way to explicitly extract a representation from intermediate layer activations in unsupervisedly pre-trained generative models. Additionally, layer activations alone do not suffice to reach the full power of learned representations hidden in neural network models, as shown in recent works (Mu et al., 2020) that incorporating additional gradients-based features into representation leads to substantial improvement over solely using activations-based features.

In light of these constraints, we are interested in the question: **is there a principled method for representation extraction beyond layer activations?** In this work, we turn to the kernel view of neural networks. Recently, initiated by the Neural Tangent Kernel (NTK) (Jacot et al., 2018) work, there have been growing interests in the kernel interpretation of neural networks. It was shown that neural networks in the infinite width regime are reduced to kernel regression with the induced NTK. Our key intuition is that, the kernel machine induced by the neural network provides a powerful and principled way of investigating the non-linear feature transformation in neural networks using the

*linear* feature space of the kernel. Kernel machines provide drastically different representations than layer activations, where the knowledge of a neural network is instantiated by the induced kernel function over data points.

In this work, we propose to make use of the linear feature space of the kernel, associated with the pre-trained neural network model, as the data representation of interest. To this end, we made novel contributions on both theoretical and empirical side, as summarized below.

- We propose Neural Fisher Kernel (NFK) as a unified and principled kernel formulation for neural networks models in both supervised learning and unsupervised learning settings.

- We introduce a highly efficient and scalable algorithm for low-rank kernel approximation of NFK, which allows us to obtain a compact yet informative feature embedding as the data representation.

- We validate the effectiveness of proposed approach from NFK in unsupervised learning, semi-supervised learning and supervised learning settings, showing that our method enjoys superior sample efficiency and representation quality.

## 2 PRELIMINARY AND RELATED WORKS

In this section, we present technical background and formalize the motivation. We start by introducing the notion of data representation from the perspective of kernel methods, then introduce the connections between neural network models and kernel methods.

**Notations**. Throughout this paper, we consider dataset with $N$ data examples $\mathcal{D} \equiv \{(\mathbf{x}_i, y_i)\}$, we use $p(\mathbf{x})$ to denote the probability density function for the data distribution and use $p_{\text{data}}(\mathbf{x})$ to denote the empirical data distribution from $\mathcal{D}$.

**Kernel Methods**. Kernel methods (Hofmann et al., 2008) have long been a staple of practical machine learning. At their core, a kernel method relies on a kernel function which acts as a similarity function between different data examples in some feature space. Here we consider positive definite kernels $\mathcal{K} : \mathcal{X} \times \mathcal{X} \to \mathbb{R}$ over a metric space $\mathcal{X}$ which defines a reproducing kernel Hilbert space $\mathcal{H}$ of function from $\mathcal{X}$ to $\mathbb{R}$, along with a mapping function $\varphi : \mathcal{X} \to \mathcal{H}$, such that the kernel function can be decomposed into the inner product $\mathcal{K}(\mathbf{x}, \mathbf{z}) = \langle \varphi(\mathbf{x}), \varphi(\mathbf{z}) \rangle$. Kernel methods aim to find a predictive linear function $f(\mathbf{x}) = \langle f, \varphi(\mathbf{x}) \rangle_{\mathcal{H}}$ in $\mathcal{H}$, which gives label output prediction for each data point $\mathbf{x} \in \mathcal{X}$. The kernel maps each data example $\mathbf{x} \in \mathcal{X}$ to a linear feature space $\varphi(\mathbf{x})$, which is the **data representation** of interest. Given dataset $\mathcal{D}$, the predictive model function $f$ is typically estimated via Kernel Ridge Regression (KRR), $\widehat{f} = \operatorname{argmin}_{f \in \mathcal{H}} \frac{1}{N} \sum_{i=1}^{N} (f(\mathbf{x}_i) - y_i)^2 + \lambda \|f\|_{\mathcal{H}}^2$.

**Neural Networks and Kernel Methods**. A long line of works (Neal, 1996; Williams, 1996; Roux and Bengio, 2007; Hazan and Jaakkola, 2015; Lee et al., 2018; de G. Matthews et al., 2018; Jacot et al., 2018; Chen and Xu, 2021; Geifman et al., 2020; Belkin et al., 2018; Ghorbani et al., 2020), have studied that many kernel formulations can be associated to neural networks, while most of them correspond to neural network where being fixed kernels (e.g. Laplace kernel, Gaussian kernel) or only the last layer is trained, e.g., Conjugate Kernel (CK) (Daniely et al., 2016), also called as NNGP kernel (Lee et al., 2018). On the other hand, Neural Tangent Kernel (NTK) (Jacot et al., 2018) is a fundamentally different formulation corresponding to training the entire infinitely wide neural network models. Let $f(\boldsymbol{\theta}; \mathbf{x})$ denote a neural network function with parameters $\boldsymbol{\theta}$, then the empirical NTK is defined as $\mathcal{K}_{\text{ntk}}(\mathbf{x}, \mathbf{z}) = \langle \nabla_{\boldsymbol{\theta}} f(\boldsymbol{\theta}; \mathbf{x}), \nabla_{\boldsymbol{\theta}} f(\boldsymbol{\theta}; \mathbf{z}) \rangle$. (Jacot et al., 2018; Lee et al., 2018) showed that under the so-called NTK parametrization and other proper assumptios, the function $f(\mathbf{x}; \boldsymbol{\theta})$ learned by training the neural network model with gradient descent is equivalent to the function estimated via ridgeless KRR using $\mathcal{K}_{\text{ntk}}$ as the kernel. For finite-width neural networks, by taking first-order Taylor expansion of funnction $f$ around the $\boldsymbol{\theta}$, kernel regression under $\mathcal{K}_{\text{ntk}}$ can be seen as linearized neural network model at parameter $\boldsymbol{\theta}$, suggesting that pre-trained neural network models can also be studied and approximated from the perspective of kernel methods.

**Fisher Kernel**. The Fisher Kernel (FK) is first introduced in the seminal work (Jaakkola and Haussler, 1998). Given a probabilistic generative model $p_{\boldsymbol{\theta}}(x)$, the Fisher kernel is defined as: $\mathcal{K}_{\text{fisher}}(\mathbf{x}, \mathbf{z}) = \nabla_{\boldsymbol{\theta}} \log p_{\boldsymbol{\theta}}(\mathbf{x})^{\top} \mathcal{I}^{-1} \nabla_{\boldsymbol{\theta}} \log p_{\boldsymbol{\theta}}(\mathbf{z}) = U_{\mathbf{x}}^{\top} \mathcal{I}^{-1} U_{\mathbf{z}}$ where $U_{\mathbf{x}} = \nabla_{\boldsymbol{\theta}} \log p_{\boldsymbol{\theta}}(\mathbf{x})$ is the so-called Fisher score and $\mathcal{I}$ is the Fisher Information Matrix (FIM) defined as the covariance of the Fisher score: $\mathcal{I} = \mathbb{E}_{\mathbf{x} \sim p_{\boldsymbol{\theta}}(\mathbf{x})} \nabla_{\boldsymbol{\theta}} \log p_{\boldsymbol{\theta}}(\mathbf{x}) \nabla_{\boldsymbol{\theta}} \log p_{\boldsymbol{\theta}}(\mathbf{x})^{\top}$. Then the Fisher vector is defined as $V_{\mathbf{x}} = \mathcal{I}^{-\frac{1}{2}} \nabla_{\boldsymbol{\theta}} \log p_{\boldsymbol{\theta}}(\mathbf{x}) = \mathcal{I}^{-\frac{1}{2}} U_{\mathbf{x}}$. One can utilize the Fisher Score as a mapping from the

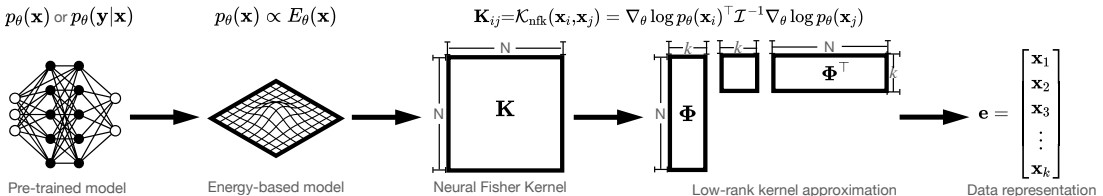

Figure 1: Overview of our proposed approach. Given a pre-trained neural network model, which can be either an unsupervised generative model $p_\theta(\mathbf{x})$ (e.g. GANs, VAEs), or a supervised learning model $p_\theta(\mathbf{y}|\mathbf{x})$, we aim to extract a compact yet informative representation from it. By reinterpreting various families of models as energy-based models (EBM), we introduce Neural Fisher Kernel (NFK) $\mathcal{K}_{\text{nfk}}$ as a principled and unified kernel formulation for neural network models (Section. 3.1). We introduce a highly efficient and scalable kernel approximation algorithm (Section. 3.2) to obtain the low-dimensional feature embedding $\mathbf{e}_\mathbf{x}$, which serves as the extracted data representation from NFK.

data space $\mathcal{X}$ to parameter space $\Theta$, and obtain representations that are linearized. As proposed in (Jaakkola and Haussler, 1998; Perronnin and Dance, 2007), the Fisher vector $V_\mathbf{x}$ can be used as the feature representation derived from probabilistic generative models, which was shown to be superior to hand-crafted visual descriptors in a variety of computer vision tasks.

**Generative Models** In this work, we consider a variety of representative deep generative models, including generative adversarial networks (GANs) (Goodfellow et al., 2014), variational auto-encoders (VAEs) (Kingma and Welling, 2014), as well we normalizing flow models (Dinh et al., 2015) and auto-regressive models (van den Oord et al., 2016). Please refer to (Salakhutdinov, 2014) for more technical details on generative models.

# 3 LEARNING REPRESENTATION FROM NEURAL FISHER KERNEL

We aim to propose a general and efficient method for extracting high-quality representation from pre-trained neural network models. As formalized in previous section, we can describe the outline of our proposed approach as: given a pre-trained neural network model $f(\mathbf{x}; \boldsymbol{\theta})$ (either unsupervised generative model $p(\mathbf{x}; \boldsymbol{\theta})$ or supervised learning model $p(\mathbf{y} \mid \mathbf{x}; \boldsymbol{\theta})$), with pre-trained weights $\boldsymbol{\theta}$, we adopt the kernel formulation $\mathcal{K}_f$ induced by model $f(\mathbf{x}; \boldsymbol{\theta})$ and make use of the associated linear feature embedding $\varphi(\mathbf{x})$ of the kernel $\mathcal{K}_f$ as the feature representation of data $\mathbf{x}$. We present an overview introduction to illustrate our approach in Figure. 1.

At this point, however, there exist both theoretical difficulties and practical challenges which impede a straightforward application of our proposed approach. On the theoretical side, the NTK theory is only developed in supervised learning setting, and its extension to unsupervised learning is not established yet. Though Fisher kernel is immediately applicable in unsupervised learning setting, deriving Fisher vector from supervised learning model $p(\mathbf{y} \mid \mathbf{x}; \boldsymbol{\theta})$ can be tricky, which needs the log-density estimation of *marginal* distribution $p_\theta(\mathbf{x})$ from $p(\mathbf{y} \mid \mathbf{x}; \boldsymbol{\theta})$. Note that it is a drastically different problem from previous works (Achille et al., 2019) where Fisher kernel is applied to the joint distribution over $p(\mathbf{x}, \mathbf{y})$. On the practical efficiency side, the dimensionality of the feature space associated with NTK or FK is same as the number of model parameters $|\boldsymbol{\theta}|$, which poses unmanageably high time and space complexity when it comes to modern large-scale neural network models. Additionally, the size of the NTK scales quadratically with the number of classes in multi-class supervised learning setting, which gives rise to more efficiency concerns.

To address the kernel formulation issue, we propose Neural Fisher Kernel (NFK) in Sec. 3.1 as a unified kernel for both supervised and unsupervised learning models. To tackle the efficiency challenge, we investigate the structural properties of the proposed NFK and propose a highly scalable low-rank kernel approximation algorithm in Sec. 3.2 to extract compact low-dimensional feature representation from NFK.

## 3.1 NEURAL FISHER KERNEL

In this section, we propose Neural Fisher Kernel (NFK) as a principled and general kernel formulation for neural network models. The key intuition is that we can extend classical Fisher kernel theory to unify the procedure of deriving Fisher vector from supervised learning models and unsupervised learning models by using Energy-based Model (EBM) formulation.

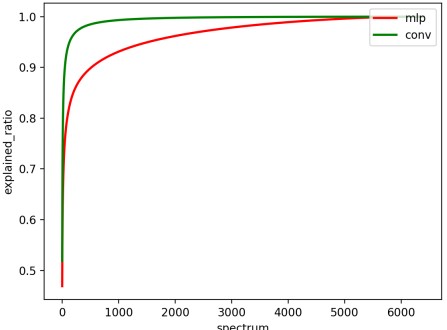 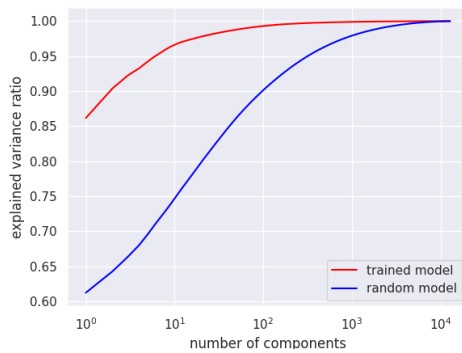

Figure 2: **Left**: The spectrum structure of NFKs from a CNN (green) and a MLP (red), trained on MNIST binary classification task. The NFK of CNN concentrates on fewer eigen-modes compared to the MLP. **Right**: The low-rankness of the NFK on a DCGAN trained on MNIST. For a trained model, the first 100 principle components of the Fisher Vector matrix explain 99.5% of all variances. An untrained model with the same architecture on the other hand, demonstrates a much lower degree of low-rankness.

### 3.1.1 UNSUPERVISED NFK

We consider unsupervised probabilistic generative models $p_{\boldsymbol{\theta}}(\mathbf{x}) = p(\mathbf{x}; \boldsymbol{\theta})$ here. Our proposed NFK formulation can be applied to all generative models with tractable evaluation (or approximation) of $\nabla_{\boldsymbol{\theta}} \log p_{\boldsymbol{\theta}}(\mathbf{x})$.

**GANs**. We consider the EBM formulation of GANs (Dai et al., 2017; Zhai et al., 2019; Che et al., 2020). Given pre-trained GAN model, we use $D(\mathbf{x}; \boldsymbol{\theta})$ to denote the output of the discriminator $D$, and use $G(\mathbf{h})$ to denote the output of generator $G$ given latent code $\mathbf{h} \sim p(\mathbf{h})$. As a brief recap, GANs can be interpreted as an implementation of EBM training with a variational distribution, where we have the energy-function $E(\mathbf{x}; \boldsymbol{\theta}) = -D(\mathbf{x}; \boldsymbol{\theta})$. Please refer to (Zhai et al., 2019; Che et al., 2020) for more details. Thus we have the unnormalized density function $p_{\boldsymbol{\theta}}(\mathbf{x}) \propto e^{-E(\mathbf{x}; \boldsymbol{\theta})}$ given by the GAN model. Following (Zhai et al., 2019), we can then derive the Fisher kernel $\mathcal{K}_{\mathrm{nfk}}$ and Fisher vector from standard GANs as shown below:

$$
\begin{aligned}
\mathcal{K}_{\mathrm{nfk}}(\mathbf{x}, \mathbf{z}) &= \langle V_{\mathbf{x}}, V_{\mathbf{z}} \rangle \qquad V_{\mathbf{x}} = (\mathtt{diag}(\mathcal{I})^{-\frac{1}{2}}) U_{\mathbf{x}} \\
U_{\mathbf{x}} &= \nabla_{\boldsymbol{\theta}} D(\mathbf{x}; \boldsymbol{\theta}) - \mathbb{E}_{\mathbf{h} \sim p(\mathbf{h})} \nabla_{\boldsymbol{\theta}} D(G(\mathbf{h}); \boldsymbol{\theta})
\end{aligned}
\tag{1}
$$

where $\mathbf{x}, \mathbf{z} \in \mathcal{X}$, $\mathcal{I} = \mathbb{E}_{\mathbf{h} \sim p(\mathbf{h})} \left[ U_{G(\mathbf{h})} U_{G(\mathbf{h})}^{\top} \right]$. Note that we use diagonal approximation of FIM throughout this work for the consideration of scalability.

**VAEs**. Given a VAE model pre-trained via maximizing the variational lower-bound ELBO $\mathcal{L}_{\mathrm{ELBO}}(\mathbf{x}) \equiv \mathbb{E}_{q(\mathbf{h}|\mathbf{x})} \left[ \log \frac{p(\mathbf{x}, \mathbf{h})}{q(\mathbf{h}|\mathbf{x})} \right]$, we can approximate the marginal log-likelihood $\log p_{\boldsymbol{\theta}}(\boldsymbol{\theta})$ by evaluating $\mathcal{L}_{\mathrm{ELBO}}(\mathbf{x})$ via Monte-Carlo estimations or importance sampling techniques (Burda et al., 2016). Thus we have our NFK formulation as

$$
\begin{aligned}
\mathcal{K}_{\mathrm{nfk}}(\mathbf{x}, \mathbf{z}) &= \langle V_{\mathbf{x}}, V_{\mathbf{z}} \rangle \qquad V_{\mathbf{x}} = (\mathtt{diag}(\mathcal{I})^{-\frac{1}{2}}) U_{\mathbf{x}} \\
U_{\mathbf{x}} &\approx \nabla_{\boldsymbol{\theta}} \mathcal{L}_{\mathrm{ELBO}}(\mathbf{x})
\end{aligned}
\tag{2}
$$

where $\mathbf{x}, \mathbf{z} \in \mathcal{X}$, $\mathcal{I} = \mathbb{E}_{\mathbf{x} \sim p_{\boldsymbol{\theta}}(\mathbf{x})} \left[ U_{\mathbf{x}} U_{\mathbf{x}}^{\top} \right]$.

**Flow-based Models, Auto-Regressive Models**. For generative models with explicit exact data density modeling $p_{\boldsymbol{\theta}}(\mathbf{x})$, we can simply apply the classical Fisher kernel formulation in Sec. 2.

### 3.1.2 SUPERVISED NFK

In the supervised learning setting, we consider conditional probabilistic models $p_{\boldsymbol{\theta}}(\mathbf{y} \mid \mathbf{x}) = p(\mathbf{y} \mid \mathbf{x}; \boldsymbol{\theta})$. In particular, we focus on classification problems where the conditional probability is parameterized by a softmax function over the logits output $f(\mathbf{x}; \boldsymbol{\theta})$: $p_{\boldsymbol{\theta}}(\mathbf{y} \mid \mathbf{x}) = \exp(f^{\mathbf{y}}(\mathbf{x}; \boldsymbol{\theta})) / \sum_{\mathbf{y}} \exp(f^{\mathbf{y}}(\mathbf{x}; \boldsymbol{\theta}))$, where $\mathbf{y}$ is a discrete label and $f^{\mathbf{y}}(\mathbf{x}; \boldsymbol{\theta})$ denotes $\mathbf{y}$-th logit output. We then borrow the idea from JEM (Grathwohl et al., 2020) and write out a joint energy function term over $(\mathbf{x}, \mathbf{y})$ as $E(\mathbf{x}, \mathbf{y}; \boldsymbol{\theta}) = -f^{\mathbf{y}}(\mathbf{x}; \boldsymbol{\theta})$. It is easy to see that joint energy yields exactly the same conditional probability, at the same time leading to a free energy function:

---

**Algorithm 1** Baseline method: compute low-rank NFK feature embedding

---

**Input** dataset $\mathcal{D}$; pre-train NN model $f(\mathbf{x}; \boldsymbol{\theta})$; NFK feature dimensionality $k$; test data input $\mathbf{x}^\star$
**Output** low-rank NFK feature embedding $\mathbf{e}_{\mathrm{nfk}}(\mathbf{x}^*)$

1: compute Fisher vector for all data examples $\mathbf{V} = [V_{\mathbf{x}_i}] \in \mathbb{R}^{N \times |\boldsymbol{\theta}|}$
2: compute kernel Gram matrix $\mathbf{K} = \mathbf{V}\mathbf{V}^\top \in \mathbb{R}^{N \times N}$
3: compute truncated eigen-decomposition $\mathbf{K} = \boldsymbol{\Phi}\mathtt{diag}(\Lambda)\boldsymbol{\Phi}^\top, \boldsymbol{\Phi} \in \mathbb{R}^{N \times k}$
4: kernel function evaluations between $\mathbf{x}^*$ and all data examples $\mathcal{K}(\mathbf{x}^\star, \mathbf{X}) \equiv [\mathcal{K}(\mathbf{x}^\star, \mathbf{x}_j)]_{j=1}^N$
5: obtain $\mathbf{e}_{\mathrm{nfk}}(\mathbf{x}^*) \in \mathbb{R}^k$ via Eq. 5 and Eq. 4

---

$E(\mathbf{x}; \boldsymbol{\theta}) = -\log \sum_{\mathbf{y}} \exp(f^{\mathbf{y}}(\mathbf{x}; \boldsymbol{\theta}))$. It essentially reframes a conditional distribution over $\mathbf{y}$ given $\mathbf{x}$ to an induced unconditional distribution over $\mathbf{x}$, while maintaining the same conditional probability $p_{\boldsymbol{\theta}}(\mathbf{y} \mid \mathbf{x})$. This allows us to write out the NFK formulation as:

$$\mathcal{K}_{\mathrm{nfk}}(\mathbf{x}, \mathbf{z}) = \langle V_{\mathbf{x}}, V_{\mathbf{z}} \rangle \quad V_{\mathbf{x}} = (\mathtt{diag}(\mathcal{I})^{-\frac{1}{2}}) U_{\mathbf{x}}$$
$$U_{\mathbf{x}} = \sum_{\mathbf{y}} p_{\boldsymbol{\theta}}(\mathbf{y} \mid \mathbf{x}) \nabla_{\boldsymbol{\theta}} f^{\mathbf{y}}(\mathbf{x}; \boldsymbol{\theta}) - \mathbb{E}_{\mathbf{x}' \sim p_{\boldsymbol{\theta}}(\mathbf{x}')} \sum_{\mathbf{y}} p_{\boldsymbol{\theta}}(\mathbf{y} \mid \mathbf{x}) \nabla_{\boldsymbol{\theta}} f^{\mathbf{y}}(\mathbf{x}'; \boldsymbol{\theta}) \tag{3}$$

where $\mathcal{I} = \mathbb{E}_{\mathbf{x} \sim p_{\boldsymbol{\theta}}(\mathbf{x})} [U_{\mathbf{x}} U_{\mathbf{x}}^\top]$, and $p_{\boldsymbol{\theta}}(\mathbf{x})$ is the normalized density corresponding to the free energy $E_{\boldsymbol{\theta}}$, which could be sampled from via Markov chain Monte Carlo (MCMC) algorithm. In this work, we use empirical data distribution as practical approximation.

## 3.2 NFK with Low-Rank Approximation

Fisher vector $V_{\mathbf{x}}$ is the linear feature embedding $\varphi(\mathbf{x})$ given by NFK $\mathcal{K}_{\mathrm{nfk}}(\mathbf{x}, \mathbf{z}) = \langle V_{\mathbf{x}}, V_{\mathbf{z}} \rangle$ for neural network model $f(\mathbf{x}; \boldsymbol{\theta})$. However, straightforward application of NFK by using $V_x$ as feature representation suffers from scalability issue, since $V_{\mathbf{x}} \in \mathbb{R}^{|\boldsymbol{\theta}|}$ shares same dimensionality as the number of parameters $|\boldsymbol{\theta}|$. It is with that in mind that $|\boldsymbol{\theta}|$ can be tremendously large considering the scale of modern neural networks, it is unfortunately infeasible to directly leverage $V_{\mathbf{x}}$ as feature representation.

**Low-Rank Structure of NFK**. Motivated by the *Manifold Hypothesis of Data* that it is widely believed that real world high dimensional data lives in a low dimensional manifold (Roweis and Saul, 2000; Rifai et al., 2011a;b), we investigate the structure of NFKs and present empirical evidence that *NFKs of good models have low-rank spectral structure*. Firstly, we start by examining supervised learning models. We study the spectrum structure of the empirical NFK of trained neural networks with different architectures. We trained a LeNet-5 (LeCun et al., 1998) CNN and a 3-layer MLP network by minimizing binary cross entropy loss, and then compute the eigen-decomposition of the NFK Gram matrix. We show the explained ratio plot in Figure 2. We see that the spectrum of CNN NTK concentrates on fewer large eigenvalues, thus exhibiting a lower effective-rank structure compared to the MLP, which can be explained by the fact that CNN has better model inductive bias for image data domain. For unsupervised learning models, we trained a small unconditional DCGAN (Radford et al., 2016) model on MNIST dataset. We compare the results of a fully trained model against a randomly initialized model in Fig. 2 (note the logarithm scale of the $x$-axis). Remarkably, the trained model demonstrates an extreme degree of low-rankness that top 100 principle components explain over $99.5\%$ of the overall variance, where 100 is two orders of magnitude smaller than both number of examples and number of parameters in the discriminator. We include more experimental results and discussions in appendix due to the space constraints.

**Efficient Low-Rank Approximation of NFK**. The theoretical insights and empirical evidence presented above hint at a natural solution to address the challenge of high-dimensionality of $V_{\mathbf{x}} \in \mathbb{R}^{|\boldsymbol{\theta}|}$: we can turn to seek a low-rank approximation to the NFK. According to the Mercer's theorem (Mercer, 1909), for positive definite kernel $\mathcal{K}(\mathbf{x}, \mathbf{z}) = \langle \varphi(\mathbf{x}), \varphi(\mathbf{z}) \rangle$ we have $\mathcal{K}(\mathbf{x}, \mathbf{z}) = \sum_{i=1}^\infty \lambda_i \phi_i(\mathbf{x}) \phi_i(\mathbf{z})$, $\mathbf{x}, \mathbf{z} \in \mathcal{X}$, where $\{(\lambda_i, \phi_i)\}$ are the eigenvalues and eigenfunctions of the kernel $\mathcal{K}$, with respect to the integral operator $\int p(\mathbf{z}) \mathcal{K}(\mathbf{x}, \mathbf{z}) \phi_i(\mathbf{z}) \, \mathrm{d}\mathbf{z} = \lambda_i \phi_i(\mathbf{x})$. The linear feature embedding representation $\varphi(\mathbf{x})$ can thus be constructed from the orthonormal eigenbasis $\{(\lambda_i, \phi_i)\}$ as $\varphi(\mathbf{x}) \equiv [\sqrt{\lambda_1}\phi_1(\mathbf{x}), \sqrt{\lambda_2}\phi_2(\mathbf{x}), \ldots] \equiv [\sqrt{\lambda_i}\phi_i(\mathbf{x})]$, $i = 1, \ldots, \infty$. To obtain a low-rank approximation, we only keep top-$k$ largest eigen-basis $\{(\lambda_i, \phi_i)\}$ ordered by corresponding eigenvalues $\lambda_i$ to form the low-rank $k$-dimensional feature embedding $\mathbf{e}(\mathbf{x}) \in \mathbb{R}^k$, $k \ll N, k \ll |\boldsymbol{\theta}|$

$$\mathbf{e}(\mathbf{x}) \equiv \left[\sqrt{\lambda_1}\phi_1(\mathbf{x}), \sqrt{\lambda_2}\phi_2(\mathbf{x}), \ldots \sqrt{\lambda_k}\phi_k(\mathbf{x})\right] \equiv \left[\sqrt{\lambda_i}\phi_i(\mathbf{x})\right], \quad i = 1, \ldots, k \tag{4}$$

---

**Algorithm 2** Our proposed method: compute low-rank NFK feature embedding

---
1: $\mathbf{V} = \boldsymbol{\Phi}\texttt{diag}(\Sigma)\mathbf{P}^\top, \mathbf{P} \in \mathbb{R}^{|\boldsymbol{\theta}|\times k}$ using power iteration methods via JVP/VJP evaluations
2: compute $\mathcal{K}\left(\mathbf{x}, \mathbf{X}\right)^\top \boldsymbol{\Phi}_i \approx V_{\mathbf{x}}\texttt{diag}(\Sigma_i)\mathbf{P}_i$ via JVP evaluation
3: obtain $\mathbf{e}_{\text{nfk}}(\mathbf{x}^*) \in \mathbb{R}^k$ via Eq. 5 and Eq. 4

---

By applying our proposed NFK formulation $\mathcal{K}_{\text{nfk}}$ to pre-trained neural network model $f(\mathbf{x}; \boldsymbol{\theta})$, we can obtain a compact low-dimensional feature representation $\mathbf{e}_{\text{nfk}}(\mathbf{x}) \in \mathbb{R}^k$ in this way. We call it the **low-rank NFK feature embedding**.

We then illustrate how to estimate the eigenvalues and eigenfunctions of NFK $\mathcal{K}_{\text{nfk}}$ from data. Given dataset $\mathcal{D}$, the Gram matrix $\mathbf{K} \in \mathbb{R}^{N\times N}$ of kernel $\mathcal{K}$ is defined as $\mathbf{K}(\mathbf{x}_i, \mathbf{x}_j) = \mathcal{K}(\mathbf{x}_i, \mathbf{x}_j)$. We use $\mathbf{X} \equiv [\mathbf{x}_i]_{i=1}^N$ to denote the matrix of all data examples, and use $\phi_i(\mathbf{X}) \in \mathbb{R}^N$ to denote the concatenated vector of evaluating $i$-th eigenfunction $\phi_i$ at all data examples. Then by performing eigen-decomposition of the Gram matrix $\mathbf{K} = \boldsymbol{\Phi}\texttt{diag}(\Lambda)\boldsymbol{\Phi}^\top$, the $i$-th eigenvector $\boldsymbol{\Phi}_i \in \mathbb{R}^N$ and eigenvalue $\Lambda_i$ can be seen as unbiased estimation of the $i$-th eigenfunction $\phi_i$ and eigenvalue $\lambda_i$ of the kernel $\mathcal{K}$, evaluated at training data examples $\mathbf{X}$, $\phi_i(\mathbf{X}) \approx \sqrt{N}\boldsymbol{\Phi}_i, \lambda_i \approx \frac{1}{N}\Lambda_i$. Based on these estimations, we can thus approximate the eigenfunction $\phi_i$ via the integral operator by Monte-Carlo estimation with empirical data distribution,

$$\lambda_i \phi_i(\mathbf{x}) = \int p(\mathbf{z})\mathcal{K}(\mathbf{x}, \mathbf{z})\phi_i(\mathbf{z})\,\mathrm{d}\mathbf{z} \approx \mathbb{E}_{\mathbf{x}_j \in p_{\text{data}}}\mathcal{K}(\mathbf{x}, \mathbf{x}_j)\phi_j(\mathbf{x}_j) \approx \frac{1}{N}\sum_{j=1}^N \mathcal{K}(\mathbf{x}, \mathbf{x}_j)\boldsymbol{\Phi}_{ji} \quad (5)$$

Given new test data example $\mathbf{x}^\star$, we can now approximate the eigenfunction evaluation $\phi_i(\mathbf{x}^\star)$ by the projection of kernel function evaluation results centered on training data examples $\mathcal{K}(\mathbf{x}^\star, \mathbf{X}) \equiv [\mathcal{K}(\mathbf{x}^\star, \mathbf{x}_j)]_{j=1}^N$ onto the $i$-th eigenvector $\boldsymbol{\Phi}_i$ of kernel Gram matrix $\mathbf{K}$. We adopt this method as the baseline approach for low-rank approximation, and present the baseline algorithm description in Alg. 1.

However, due to the fact that it demands explicit computation and manipulation of the Fisher vector matrix $\mathbf{V} \in \mathbb{R}^{N\times|\boldsymbol{\theta}|}$ and the Gram matrix $\mathbf{K} \in \mathbb{R}^{N\times N}$ in Alg. 1, straightforward application of the baseline approach, as well as other off-the-shelf classical kernel approximation (Williams and Seeger, 2000; Rahimi and Recht, 2007) and SVD methods (Halko et al., 2011), are practically infeasible to scale to larger-scale machine learning settings, where both the number of data examples $N$ and the number of model parameters $|\boldsymbol{\theta}|$ can be extremely large.

To tackle the posed scalability issue, we propose a novel highly efficient and scalable algorithm for computing low-rank approximation of NFK. Given dataset $\mathcal{D}$ and model $f(\mathbf{x}; \boldsymbol{\theta})$, We aim to compute the truncated SVD of the Fisher vector matrix $\mathbf{V} = \boldsymbol{\Phi}\texttt{diag}(\Sigma)\mathbf{P}^\top, \mathbf{P} \in \mathbb{R}^{|\boldsymbol{\theta}|\times k}$. Based on the idea of power methods (Golub and Van der Vorst, 2000; Bathe, 1971) for finding leading top eigenvectors, we start from a random vector $\mathbf{v}_0$ and iteratively construct the sequence $\mathbf{v}_{t+1} = \frac{\mathbf{V}\mathbf{V}^\top \mathbf{v}_t}{\|\mathbf{V}\mathbf{V}^\top \mathbf{v}_t\|}$. By leveraging the special structure of $\mathbf{V}$ that it can be obtained from the Jacobian matrix $J_{\boldsymbol{\theta}}(\mathbf{X}) \in \mathbb{R}^{N\times|\boldsymbol{\theta}|}$ up to element-wise linear transformation under the NFK formulation in Sec. 3, we can decompose each iterative step into a Jacobian Vector Product (JVP) and a Vector Jacobian Product (VJP). With modern automatic-differentiation techniques, we can evaluate both JVP and VJP efficiently, which only requires the same order of computational costs of one vanilla backward-pass and forward-pass of neural networks respectively. With computed truncated SVD results, we can approximate the projection term in Eq. 5 by $\mathcal{K}\left(\mathbf{x}, \mathbf{X}\right)^\top \boldsymbol{\Phi}_i = V_{\mathbf{x}}\mathbf{V}^\top \boldsymbol{\Phi}_i \approx V_{\mathbf{x}}\texttt{diag}(\Sigma_i)\mathbf{P}_i$, which is again in the JVP form so that we can pre-compute and store the truncated SVD results and evaluate the eigenfunction of any test data example via one efficient JVP forward-pass. We describe our proposed algorithm briefly in Alg. 2.

## 4 EXPERIMENTS

In this section, we evaluate NFK in the following settings. We first evaluate the proposed low-rank kernel approximation algorithm (Sec. 3.2), in terms of both approximation accuracy and running time efficiency. Next, we evaluate NFK on various representation learning tasks in both supervised, semi-supervised and unsupervised learning settings.

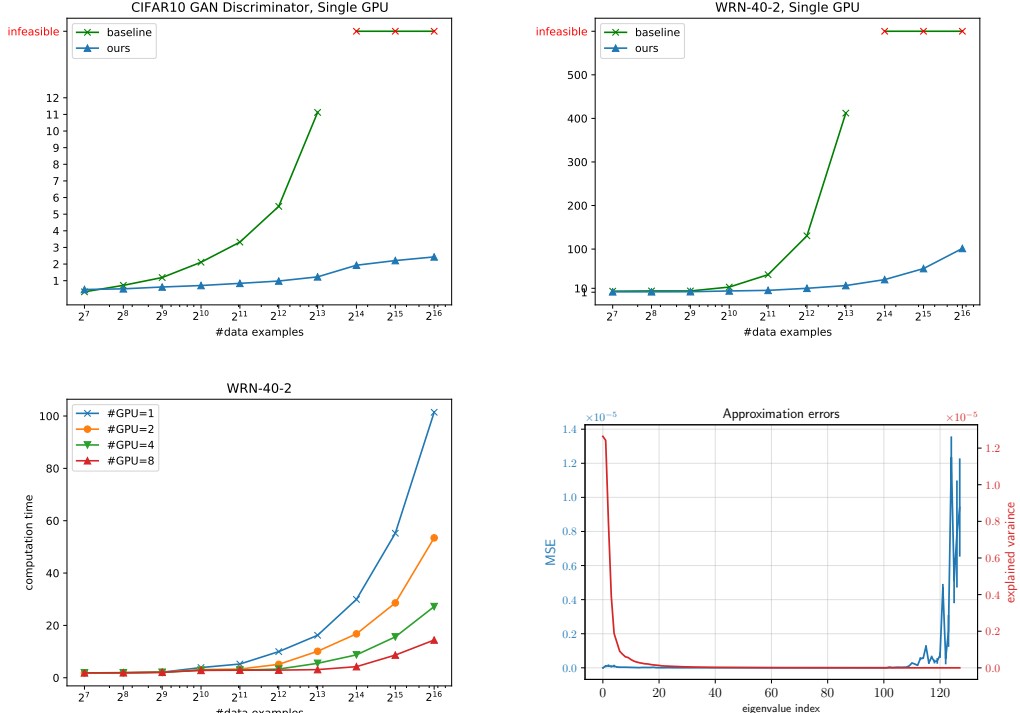

Figure 3: **Top row**: Running time efficiency evaluation for truncated SVD algorithm on single GPU. We vary the number of data examples used, shown in $x$-axis. $y$-axis denotes the wall-clock running time (in seconds). Red crosses mark the cases when it is no longer possible for the baseline method to obtain the results in an affordable waiting time and memory consumption. **Bottom left**: Running time costs with different number of GPUs used in our distributed SVD implementation. **Bottom right**: Approximation errors (in blue) of our proposed implementation for each eigenmode (in descending order of eigenvalues), v.s. the explained variance (in red). Best viewed in color.

## 4.1 QUALITY AND EFFICIENCY OF LOW-RANK NFK APPROXIMATIONS

We implement our proposed low-rank kernel approximation algorithm in Jax (Bradbury et al., 2018) with distributed multi-GPU parallel computation support. For the baseline methods for comparison, we first compute the full kernel Gram matrix using the `neural-tangents` (Novak et al., 2020) library, and then use `sklearn.decomposition.TruncatedSVD` to obtain the truncated SVD results. All model and algorithm hyper-parameters are included in the Appendix.

**Computational Costs**. We start by comparing running time costs of computing top NFK eigenvectors via truncated SVD. We use two models for the comparison, a DCGAN-like GAN model in (Zhai et al., 2019) and a Wide ResNet (WRN) with $40$ layers and $2$ times wider than original network (denoted as WRN-$40$-2). Please see appendix for the detailed description of hyper-parameters. We observed that our proposed algorithm could achieve nearly linear time scaling, while the baseline method would not be able to handle more than $2^{14}$ data examples as the memory usage and time complexity are too high to afford. We also see in Fig. 3 that by utilizing multi-GPU parallelism, we achieved further speed-up which scales almost linearly with the number of GPUs. We emphasize that given the number of desired eigenvectors, the time complexity of our method scales linearly with the number of data examples and the demanded memory usage remains constant with adequate data batch size, since explicit computation and storage of the full kernel matrix is never needed.

**Approximation accuracy**. We investigate the approximation error of our proposed low-rank approximation method. Since we did not introduce any additional approximations, our method shares the same approximation error bound with the existing randomized SVD algorithm (Martinsson and Tropp, 2020; Halko et al., 2011) and would only expect differences compared to the baseline randomized SVD algorithm up to numerical errors. To evaluate the quality of the low-rank kernel approximation, we use LeNet-5 and compute its full NFK Gram matrix on MNIST dataset. Please see appendix for detailed hyper-parameter setups. We show in Fig. 3 the approximation errors of top-$128$ eigenvalues along with corresponding explained variances. We obtain less than $1e-8$ absolute error and less than $1e-7$ relative error in top eigen-modes which explains most of the data.

Table 1: CIFAR-10 accuracies of linear evaluation on top of representations learned with unsupervised and self-supervised methods. NFK-128d denotes the 128 dimensional embeddings from the low-rank approximation of the NFK (ie AFV). Remarkably, we can use 128 dimensions to exactly recover the performance of the 5.9M dimensional Fisher Vectors.

| Model | Acc | Category | #Features |
|---|---|---|---|
| Examplar CNN (Dosovitskiy et al., 2015) | 84.3 | Unsupervised | - |
| BiGAN (Mu et al., 2020) | 70.5 | Unsupervised | - |
| RotNet Linear (Gidaris et al., 2018) | 81.8 | Self-Supervised | $\sim 25K$ |
| AET Linear (Zhang et al., 2019) | 83.3 | Self-Supervised | $\sim 25K$ |
| VAE(Mu et al., 2020) | 61.5 | Unsupervised | - |
| VAE-NFK-128d (ours) | 63.2 | Unsupervised | **128** |
| VAE-NFK-256d (ours) | 68.7 | Unsupervised | **256** |
| GAN-Supervised | 92.7 | Supervised | - |
| GAN-Activations | 65.3 | Unsupervised | - |
| GAN-AFV (Zhai et al., 2019) | 89.1 | Unsupervised | 5.9M |
| GAN-AFV (re-implementation) (Zhai et al., 2019) | 89.8 | Unsupervised | 5.9M |
| GAN-NFK-128d (ours) | **89.8** | Unsupervised | **128** |
| GAN-NFK-256d (ours) | **89.8** | Unsupervised | **256** |

## 4.2 LOW-RANK NFK EMBEDDING AS DATA REPRESENTATIONS

In this section we evaluate NFK to answer the following: **Q1**. In line with the question raised in Sec. 1, how does our proposed low-rank NFK embedding differ from the intermediate layer activations for data representation? **Q2**. How does the low-rank NFK embedding compare to simply using gradients (Jacobians) as data representation? **Q3**. To what extent can the low-rank NFK embedding preserve the information in full Fisher vector? **Q4**. Does the NFK embedding representation lead to better generalization performance in terms of better sample efficiency and faster adaptation? We conduct comparative studies on different tasks to understand the NFK embedding and present empirical observations to answer these questions in following sections.

**NFK Representations from Unsupervised Generative Models**. In order to examine the effectiveness of the low-rank NFK embeddings as data representations in unsupervised learning setting, we consider GANs and VAEs as representative generative models and compute the low-rank NFK embeddings. Then we adopt the linear probing protocol by training a linear classifier on top of the obtained embeddings and report the classification performance to quantify the quality of NFK data representation. For GANs, we use the same pretrained GAN model from (Zhai et al., 2019) and reimplemented the AFV baseline. For VAEs, we follow the same model architecture proposed in (Child, 2020). We then apply the proposed truncated SVD algorithm with 10 power iterations to obtain the 256 dimensional embedding via projection. We present our results on CIFAR-10 (Krizhevsky et al., 2009a) in Table. 1. We use `GAN-NFK-128d` (`GAN-NFK-256d`) to denote the NFK embedding obtained from using top-128 (top-256) eigenvectors in our GAN model. Our VAE models (`VAE-NFK-128d` and `VAE-NFK-256d`) follow the same notations. For VAE baselines, the method proposed in (Mu et al., 2020) combines both gradients features and activations-based features into one linear model, denoted as `VAE` in the table. For GAN baselines, we first consider using intermediate layer activations only as data representation, referred to as the `GAN-Activations` model. We then consider using full Fisher vector as representation, namely using the normalized gradients w.r.t all model parameters as features, denoted as the `GAN-AFV` model as proposed in (Zhai et al., 2019). Moreover, we also compare our results against training whole neural network using data labels in a supervised learning way, denoted as `GAN-Supervised` model.

As shown in Table. 1, by contrasting against the baseline `GAN-AFV` from `GAN-Activations`, as well as validation in recent works (Zhai et al., 2019; Mu et al., 2020), gradients provide additional useful information beyond layer activations based features. However, it would be impractical to use all gradients or full Fisher vector as representation when scaling up to large-scale neural network models. For example, VAE (Child, 2020) has $\sim 40M$ parameters, it would not be possible to apply the baseline methods directly. Our proposed low-rank NFK embedding approach addressed this challenge by building low-dim vector representation from efficient kernel approximation algorithm, making it possible to utilize all model parameters' gradients information by embedding it into a low-dim vector, e.g. 256-dimensional embedding in `VAE-NFK-256d` from $\sim 40M$ parameters. As our low-rank NFK embedding is obtained by linear projections of full Fisher vectors, it naturally provides answers for **Q2** that the NFK embedding can be viewed as a compact yet informative representation containing information from all gradients. We see from Table. 1 that by using top-128 eigenvectors,

Table 2: Error rates of semi-supervised classification on CIFAR10 and SVHN, varying labels from 500 to 4000. NFK-128d yields extremely competitive performance, compared to other more sophisticated baselines, Mixup (Zhang et al., 2018), VAT (Miyato et al., 2019), MeanTeacher(Tarvainen and Valpola, 2017), MixMatch (Berthelot et al., 2019), Improved GAN(Salimans et al., 2016), all are jointly learns with labels, . Also note that the architecture used by MixMatch yields a 4.13% supervised learning error rate, which is a much stronger than our supervised baseline (7.3%).

| Model | Category | CIFAR-10 | | | | SVHN | | | |
|---|---|---|---|---|---|---|---|---|---|
| | | 500 | 1000 | 2000 | 4000 | 500 | 1000 | 2000 | 4000 |
| Mixup | Joint | 36.17 | 25.72 | 18.14 | 13.15 | 29.62 | 16.79 | 10.47 | 7.96 |
| VAT | Joint | 26.11 | 18.68 | 14.40 | 11.05 | 7.44 | 5.98 | 4.85 | 4.20 |
| MeanTeacher | Joint | 47.32 | 42.01 | 17.32 | 12.17 | 6.45 | 3.82 | 3.75 | 3.51 |
| MixMatch | Joint | 9.65 | 7.75 | 7.03 | 6.24 | 3.64 | 3.27 | 3.04 | 2.89 |
| Improved GAN | Joint | - | 19.22 | 17.25 | 15.59 | 18.44 | 8.11 | 6.16 | - |
| NFK-128d (ours) | Pretrained | 20.68 | 14.77 | 13.82 | 12.95 | 8.74 | 4.47 | 3.82 | 3.19 |

Table 3: Supervised knowledge distillation results (classification accuracy on test dataset) on CIFAR10 against baseline methods KD (Hinton et al., 2015), FitNet (Romero et al., 2015), AT (Zagoruyko and Komodakis, 2017), NST (Huang and Wang, 2017), VID-I (Ahn et al., 2019), numbers are from (Ahn et al., 2019).

| | Teacher | Student | KD | FitNet | AT | NST | VID-I | NFKD (ours) |
|---|---|---|---|---|---|---|---|---|
| ACC | 94.26 | 90.72 | 91.27 | 90.64 | 91.60 | 91.16 | 91.85 | **92.42** |

the low-rank NFK embedding is able to recover the performance of full $\sim 5.9M$-dimension Fisher vector, which provides positive evidence for **Q3** that we can preserve most of the useful information in Fisher vector by taking advantage of the low-rank structure of NFK spectrum.

**NFK Representations for Semi-Supervised Learning**. We then test the low-rank NFK embeddings in the semi-supervised learning setting. Following the standard semi-supervised learning benchmark settings (Berthelot et al., 2019; Miyato et al., 2019; Laine and Aila, 2017; Sajjadi et al., 2016; Tarvainen and Valpola, 2017), we evaluate our method on CIFAR-10 (Krizhevsky et al., 2009a) and SVHN datasets (Krizhevsky et al., 2009b). We treat most of the dataset as unlabeled data and use few examples as labeled data. We use the same GAN model as the unsupervised learning setting above, and compute top-128 eigenvectors using training dataset (labeled and unlabeled) to derive the 128-dimensional NFK embedding. Then we only use the labeled data to train a linear classifier on top of the NFK embedding features, denoted as the `NFK-128d` model. We vary the number of labeled training examples and report the results in Table. 2, in comparison with other baseline methods. We see that `NFK-128d` achieves very competitive performance. On CIFAR-10, `NFK-128d` is only outperformed by the state-of-the-art semi-supervised learning algorithm MixMatch (Berthelot et al., 2019), which also uses a stronger architecture than ours. The results on SVHN are mixed though `NFK-128d` is competitive with the top performing approaches. The results demonstrated the effectiveness of NFK embeddings from unsupervised generative models in semi-supervised learning, showing promising sample efficiency for **Q4**.

**NFK Representations for Knowledge Distillation**. We next test the effectiveness of using low-rank NFK embedding for knowledge distillation in the supervised learning setting. We include more details of the distillation method in the Appendix. Our experiments are conducted on CIFAR10, with a teacher set as the `WRN-40-2` model and student being *WRN-16-1*. After training the teacher, we compute the low-rank approximation of NFK of the teacher model, using top-20 eigenvectors. We include more details about the distillation method setup in the Appendix. Our results are reported in Table. 3. We see that our method achieves superior results compared to other competitive baseline knowledge distillation methods, which mainly use the logits and activations from teacher network as distillation target.

## 5   CONCLUSIONS

In this work, we propose a novel principled approach to representation extraction from pre-trained neural network models. We introduce NFK by extending the Fisher kernel to neural networks in both unsupervised learning and superevised learning settings, and propose a novel low-rank kernel approximation algorithm, which allows us to obtain a compact feature representation in a highly efficient and scalable way.

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

## A    EXTENDED PRELIMINARIES

We extend Sec. 2 to introduce additional technical background and related work.

**Kernel methods in Deep Learning.** Popularized by the NTK work Jacot et al. (2018), there has been great interests in the deep learning community around the kernel view of neural networks. In particular, several works have studied the low-rank structure of the NTK, including (Baratin et al., 2021; Papyan, 2020; Canatar et al., 2020), which demonstrate that empirical NTK demonstrates low-rankness and that encourages better generalization theoretically. Our low-rank analysis of NFK shares a similar flavor, but generalizes across supervised and unsupervised learning settings. Besides, we make an explicit effort in proposing an efficient implementation of the low-rank approximation, and demonstrate strong empirical performances.

**Unsupervised/self supervised representation learning.** Unsupervised representation learning is an old idea in deep learning. A large body of work is dedicated to designing better learning objectives (self supervised learning), including denoising (Vincent et al., 2010), contrastive learning (Oord et al., 2018; Chen et al., 2020b; He et al., 2020), mutual information based methods (Hjelm et al., 2019; Poole et al., 2019; Zhang et al., 2020) and other "pretext tasks" Jing and Tian (2020). Our attempt falls into the same category of unsupervised representation learning, but differs in that we instead focus on effectively extracting information from a standard probabilistic model. This makes our effort orthogonal to many of the related works, and can be easily plugged into different family of models.

**Knowledge Distillation.** Knowledge distillation (KD) is generally concerned about the problem of supervising a student model with a teacher model (Hinton et al., 2015; Ba and Caruana, 2014). The general form of KD is to directly match the statistics of one or a few layers (default is the logits). Various works have studied the layer selection (Romero et al., 2015) or loss function design aspects (Ahn et al., 2019). More closely related to our work is efforts that consider the second order statistics between examples, including (Tung and Mori, 2019; Tian et al., 2020). NFKD differs in that we represent the teacher's knowledge in the kernel space, which is directly tied to the kernel interpretation of neural networks which introduces different inductive biases than layerwise representations.

**Neural Tangent Kernel.** Recent advancements in the understanding of neural networks have shed light on the connection between neural network training and kernel methods. In (Jacot et al., 2018), it is shown that one can use the Neural Tangent Kernel (NTK) to characterize the full training of a neural network using a kernel. Let $f(\boldsymbol{\theta}; \mathbf{x})$ denote a neural network function with parameters $\boldsymbol{\theta}$. The NTK is defined as follows:

$$\mathcal{K}_{\mathrm{ntk}}(\mathbf{x}, \mathbf{z}) = \mathbb{E}_{\theta \sim \mathcal{P}_{\boldsymbol{\theta}}} \langle \nabla_{\boldsymbol{\theta}} f(\boldsymbol{\theta}; \mathbf{x}), \nabla_{\boldsymbol{\theta}} f(\boldsymbol{\theta}; \mathbf{z}) \rangle. \tag{6}$$

where $\mathcal{P}_{\boldsymbol{\theta}}$ is the probability distribution of the initialization of $\boldsymbol{\theta}$. (Jacot et al., 2018) further demonstrates that in the large width regime, a neural network undergoing training under gradient descent essentially evolves as a linear model. Let $\boldsymbol{\theta}_0$ denote the parameter values at initialization. To determine how the function $f_t(\boldsymbol{\theta}_t; \mathbf{x})$ evolves, we may naively taylor expand the output around $\boldsymbol{\theta}_0$: $f_{t+1}(\boldsymbol{\theta}_{t+1}; \mathbf{x}) \approx f_t(\boldsymbol{\theta}_t; \mathbf{x}) - \eta \nabla_{\boldsymbol{\theta}_t} f_t(\boldsymbol{\theta}_t; \mathbf{x})^{\top}(\boldsymbol{\theta}_{t+1} - \boldsymbol{\theta}_t)$. As the weight updates are given by $\boldsymbol{\theta}_{t-1} - \boldsymbol{\theta}_t = -\frac{1}{N}\eta \sum_{i=1}^{m} \nabla_{\boldsymbol{\theta}_t} \mathcal{L}_t(x_i)$, hence we have $f_{t+1}(\boldsymbol{\theta}_{t+1}; \mathbf{x}) \approx f_t(\boldsymbol{\theta}_t; \mathbf{x}) - \eta \frac{1}{N} \sum_{i=1}^{N} \mathcal{K}_{\mathrm{ntk}}(\mathbf{x}, \mathbf{x}_i) \nabla_f \mathcal{L}_t(\mathbf{x}_i)$.

The significance of the NTK stems from two observations. 1) When suitably initialized, the NTK converges to a limit kernel when the width tends to infinity $\lim_{\mathrm{width}\to\infty} \mathcal{K}_{\mathrm{ntk}}(\mathbf{x}, \mathbf{z}; \boldsymbol{\theta}_0) = \mathring{\mathcal{K}}_{\mathrm{ntk}}(\mathbf{x}, \mathbf{z})$. 2) In that limit, the NTK remains frozen in its limit state throughout training.

## B    ON THE CONNECTIONS BETWEEN NFK AND NTK

In Sec 3.1, we showed that our definition of NFK in the supervised learning setting bares great similarity to the NTK. We provide more discussion here on the connections between NFK and NTK.

For the L2 regression loss function, the empirical fisher information reduces to $\mathcal{I} = \frac{1}{N} \sum_{i=1}^{N} \nabla_{\boldsymbol{\theta}} f_{\boldsymbol{\theta}}(\mathbf{x}) \nabla_{\boldsymbol{\theta}} f_{\boldsymbol{\theta}}(\mathbf{x})^{\top}$. Note that the fisher information matrix $\mathcal{I}$ is give by a covariance matrix of $J$, while the NTK matrix is defined as the Gram matrix of $J$, where $J$ is the Jacobian matrix, implying they share the same spectrum, and that the NTK and the NFK share the same eigenvectors. The addition of $\mathcal{I}^{-1}$ in the definition of $\mathcal{K}_{nfk}$ can be seen as a form of conditioning, facilitating fast convergence in all directions spanned by $J$.

---

**Algorithm 3** Our proposed method: compute low-rank NFK feature embedding

---

1: $\mathbf{V} = \boldsymbol{\Phi}\mathrm{diag}(\Sigma)\mathbf{P}^\top, \mathbf{P} \in \mathbb{R}^{|\boldsymbol{\theta}|\times k}$ via `truncated_svd(`$\mathbf{X}$`, `$f_{\boldsymbol{\theta}}$`, topk=`$K$`, kernel_type="NFK")`
2:
3: compute $\mathcal{K}\left(\mathbf{x},\mathbf{X}\right)^\top \boldsymbol{\Phi}_i \approx V_{\mathbf{x}}\mathrm{diag}(\Sigma_i)\mathbf{P}_i$ via JVP evaluation
4: obtain $\mathbf{e}_{\mathrm{nfk}}(\mathbf{x}^*) \in \mathbb{R}^k$ via Eq. 5 and Eq. 4

---

Equation 3 also has immediate connections to NTK. In NTK, the kernel $\mathbf{K}_{\mathrm{ntk}}(\mathbf{x},\bar{\mathbf{x}}) \in \mathbb{R}^{N\times N}$ is a matrix which measures the dot product of Jacobian for every pair of logits. The NFK, on the other hand, reduces the Jacobian $\nabla_{\boldsymbol{\theta}} f_{\boldsymbol{\theta}}(\mathbf{x})$ for each example $\mathbf{x}$ to a single vector of dimension $n$ (i.e., size of $\boldsymbol{\theta}$), weighted by the predicted probability of each class $p_{\boldsymbol{\theta}}(\mathbf{y}|\mathbf{x})$. The other notable difference between NFK and NTK is the subtractive and normalize factors, represented by $\mathbb{E}_{\mathbf{x}'\sim p_{\boldsymbol{\theta}}(\mathbf{x}')}\sum_{\mathbf{y}} p_{\boldsymbol{\theta}}(\mathbf{y}|\mathbf{x})\nabla_{\boldsymbol{\theta}} f_{\boldsymbol{\theta}}^{\mathbf{y}}(\mathbf{x}')$ and $\mathcal{I}$, respectively. This distinction is related to the difference between Natural Gradient Descent (Amari, 1998; Karakida and Osawa, 2020) and gradient descent. In a nutshell, our definition of NFK in the supervised learning setting can be considered as a reduced version of NTK, with proper normalization. These properties make NFK much more scalable w.r.t. the number of classes, and also less sensitive to the scale of model's parameters.

To better see this, we can define an "unnormalized" version of NFK as $\mathcal{K}_u(\mathbf{x},\bar{\mathbf{x}}) = [\sum_{\mathbf{y}} p_{\boldsymbol{\theta}}(\mathbf{y}\mid\mathbf{x})\nabla_{\boldsymbol{\theta}} f_{\boldsymbol{\theta}}^{\mathbf{y}}(\mathbf{x})]^\top \sum_{\mathbf{y}} p_{\boldsymbol{\theta}}(\mathbf{y}\mid\bar{\mathbf{x}})\nabla_{\boldsymbol{\theta}} f_{\boldsymbol{\theta}}^{\mathbf{y}}(\bar{\mathbf{x}})$. It is easy to see that $\mathcal{K}_u$ has the same rank as the original NFK $\mathcal{K}$, as $\mathcal{I}^{-1}$ is full rank by definition. We can then further rewrite it as

$$\mathcal{K}_u(\mathbf{x},\bar{\mathbf{x}}) = \sum_{\mathbf{y}}\sum_{\bar{\mathbf{y}}} p_{\boldsymbol{\theta}}(\mathbf{y}\mid\mathbf{x})p_{\boldsymbol{\theta}}(\bar{\mathbf{y}}|\bar{\mathbf{x}})\nabla_{\boldsymbol{\theta}} f_{\boldsymbol{\theta}}^{\mathbf{y}}(\mathbf{x})^\top \nabla_{\boldsymbol{\theta}} f_{\boldsymbol{\theta}}^{\bar{\mathbf{y}}}(\bar{\mathbf{x}}) = \sum_{\mathbf{y}}\sum_{\bar{\mathbf{y}}} p_{\boldsymbol{\theta}}(\mathbf{y}\mid\mathbf{x})p_{\boldsymbol{\theta}}(\bar{\mathbf{y}}\mid\bar{\mathbf{x}})\mathcal{K}_{\mathrm{ntk}}^{\mathbf{y},\bar{\mathbf{y}}}(\mathbf{x},\bar{\mathbf{x}})$$

(7)

In words, the unnormalized version of NFK can be considered as a reduction of NTK, where the weights of each element is weighte by the predicted probability for the respective class. If we further assume that the model of interest is well trained, as is often the case in knowledge distillation, we can approximate the $\mathcal{K}_u$ as $\mathcal{K}_{\mathrm{ntk}}^{\mathbf{y}^*,\bar{\mathbf{y}}^*}(\mathbf{x},\bar{\mathbf{x}})$, where $\mathbf{y}^* = \arg\max_{\mathbf{y}} p_{\boldsymbol{\theta}}(\mathbf{y}\mid\mathbf{x})$ and likewise for $\bar{\mathbf{y}}^*$. This suggests that the unnormalized NFK can roughly viewd as a downsampled version of NTK. As a result, we expect the unnormalized NFK (and hence the NFK) to exhibit similar low rank properties as demonstrated in the NTK literature.

**On the low-rank structure of NTK**. Consider the NTK Gram matrix $\mathbf{K}_{\mathrm{ntk}} \in \mathbb{R}^{N\times N}$ of some network Given the dataset $\{\mathbf{x}_i\}_{i=1}^N$ (for simplicity we assume a scalar output) and its eigen decomposition $\mathbf{K}_{ntk} = \sum_{j=1}^m \lambda_j \mathbf{u}_j \mathbf{u}_j^\top$. Let $f \in \mathbb{R}^N$ denote the concatenated outputs. Under GD in the linear regime, the outputs $f_t$ evolves according to:

$$\forall_j, \ \mathbf{u}_j^\top (f_{t+1} - f_t) \approx -\eta\lambda_j \mathbf{u}_j^\top \nabla_f \mathcal{L}.$$

(8)

The updates $f_{t+1} - f_t$ projected onto the bases of the kernel therefore converge at different speeds, determined by the eigenvalues $\{\lambda_j\}$. Intuitively, a good kernel-data alignment means that the $\nabla_f \mathcal{L}$ is spanned by a few eigenvectors with large corresponding eigenvalues, speeding up convergence and promoting generalization.

## C Neural Fisher Kernel with Low-Rank Approximation

### C.1 Neural Fisher Kernel Formulation

We provide detailed derivations of the various NFK formulations presented in Section. 3.

**NFK for Energy-based Models**. Consider an Energy-based Model (EBM) $p_{\boldsymbol{\theta}}(\mathbf{x}) = \frac{\exp(-E(\mathbf{x};\boldsymbol{\theta}))}{Z(\boldsymbol{\theta})}$, where $E(\mathbf{x})$ is the energy function parametrized by $\boldsymbol{\theta}$ and $Z(\boldsymbol{\theta}) = \int \exp(-E(\mathbf{x};\boldsymbol{\theta}))\,\mathrm{d}\mathbf{x}$ is the

---

**Algorithm 4** `truncated_svd`, Truncated SVD Algorithm for Low-rank Kernel Approximation. Comments are based on NTK for simplicity.

**Input** Dataset $\mathbf{X} \equiv \{\mathbf{x}_i\}_{i=1}^N$
**Input** Neural network model $f_{\boldsymbol{\theta}}$
**Input** Kernel type `kernel`, NFK or NTK
**Input** Low-rank embedding size $K$
**Input** Number of power iterations $L = 10$
**Input** Number of over samples $U = 10$
**Output** Truncated SVD of Jacobian $J_{\theta}(\mathbf{X}) \approx \mathbf{P}_k \Sigma_k \mathbf{Q}_k^\top$
1: $U = K + U$                ▷ Size of augmented set of vectors in power iterations
2: Draw random matrix $\boldsymbol{\Omega} \in \mathbb{R}^{N \times U}$
3: $\boldsymbol{\Omega} =$`matrix_jacobian_product`$(f_{\boldsymbol{\theta}}, \mathbf{X}, \boldsymbol{\Omega}, $`kernel`$)$     ▷ $\boldsymbol{\Omega} = J_{\boldsymbol{\theta}}(\mathbf{X})\boldsymbol{\Omega} \in \mathbb{R}^{M \times U}$
4: **for** step $= 1$ to $L$ **do**
5:      $\boldsymbol{\Omega} =$`jacobian_matrix_product`$(f_{\boldsymbol{\theta}}, \mathbf{X}, \boldsymbol{\Omega}, $`kernel`$)$     ▷ $\boldsymbol{\Omega} = J_{\boldsymbol{\theta}}^\top(\mathbf{X})\boldsymbol{\Omega} \in \mathbb{R}^{N \times U}$
6:      $\boldsymbol{\Omega} =$`matrix_jacobian_product`$(f_{\boldsymbol{\theta}}, \mathbf{X}, \boldsymbol{\Omega}, $`kernel`$)$     ▷ $\boldsymbol{\Omega} = J_{\boldsymbol{\theta}}(\mathbf{X})\boldsymbol{\Omega} \in \mathbb{R}^{M \times U}$
7:      $\boldsymbol{\Omega} =$`qr_decomposition`$(\boldsymbol{\Omega})$
8: **end for**
9: $\mathbf{B} =$`jacobian_matrix_product`$(f_{\boldsymbol{\theta}}, \mathbf{X}, \boldsymbol{\Omega}, $`kernel`$)$     ▷ $\mathbf{B} = J_{\boldsymbol{\theta}}^\top(\mathbf{X})\boldsymbol{\Omega} \in \mathbb{R}^{N \times U}$
10: $\mathbf{P}, \boldsymbol{\Sigma}, \mathbf{Q}^\top =$`svd`$(\mathbf{B}^\top)$
11: $\mathbf{P} = \boldsymbol{\Omega}\mathbf{P}$
12: Keep top rank-$K$ vectors to obtain the truncated results $\mathbf{P}_k, \boldsymbol{\Sigma}_k, \mathbf{Q}_k^\top$
13: Return $\mathbf{P}_k, \boldsymbol{\Sigma}_k, \mathbf{Q}_k^\top$

---

**Algorithm 5** `jacobian_matrix_product`

**Input** Neural network model $f_{\boldsymbol{\theta}}$
**Input** Input data $\mathbf{X} \in \mathbb{R}^{B \times D}$, where $B$ is batch size
**Input** Input matrix $\mathbf{M}$
**Input** Kernel type `kernel`, NFK or NTK
**Output** $J_{\boldsymbol{\theta}}^\top(\mathbf{X})\mathbf{M}$ for NTK, Fisher-vector-matrix-product $V_{\boldsymbol{\theta}}^\top(\mathbf{X})\mathbf{M}$ for NFK
1: `jmp_fn` $=$ `jax.vmap(jax.jvp)`
2: $\mathbf{P} =$`jmp_fn`$(f_{\boldsymbol{\theta}}, \mathbf{X}, \mathbf{M})$
3: **if** `kernel` $=$ "NFK" **then**
4:      $\mathbf{P} = \text{diag}(\mathcal{I})^{-\frac{1}{2}}(\mathbf{P} - \mathbf{Z}_{\boldsymbol{\theta}}^\top \mathbf{M})$
5: **end if**
6: Return $\mathbf{P}$

---

partition function, we could apply the Fisher kernel formulation to derive the Fisher score $U_{\mathbf{x}}$ as

$$
\begin{aligned}
U_{\mathbf{x}} = \nabla_{\boldsymbol{\theta}} \log p_{\boldsymbol{\theta}}(\mathbf{x}) &= \nabla_{\boldsymbol{\theta}} \log\left[\exp(-E(\mathbf{x}; \boldsymbol{\theta}))\right] - \nabla_{\boldsymbol{\theta}} \log Z(\boldsymbol{\theta}) \\
&= -\nabla_{\boldsymbol{\theta}} E(\mathbf{x}; \boldsymbol{\theta}) - \nabla_{\boldsymbol{\theta}} \log Z(\boldsymbol{\theta}) \\
&= -\nabla_{\boldsymbol{\theta}} E(\mathbf{x}; \boldsymbol{\theta}) - \mathbb{E}_{\mathbf{x} \sim p_{\boldsymbol{\theta}}(\mathbf{x})} \nabla_{\boldsymbol{\theta}} \log\left[\exp(-E(\mathbf{x}; \boldsymbol{\theta}))\right] \\
&= \mathbb{E}_{\mathbf{x} \sim p_{\boldsymbol{\theta}}(\mathbf{x})} \nabla_{\boldsymbol{\theta}} E(\mathbf{x}; \boldsymbol{\theta}) - \nabla_{\boldsymbol{\theta}} E(\mathbf{x}; \boldsymbol{\theta})
\end{aligned} \tag{9}
$$

Then we can obtain the FIM $\mathcal{I}$ and the Fisher vector $V_{\mathbf{x}}$ from above results, shown as below

$$
\begin{aligned}
\mathcal{I} &= \mathbb{E}_{\mathbf{x} \sim p_{\boldsymbol{\theta}}(\mathbf{x})}\left[U_{\mathbf{x}} U_{\mathbf{x}}^\top\right] \\
V_{\mathbf{x}} &= \mathcal{I}^{-\frac{1}{2}} U_{\mathbf{x}}
\end{aligned} \tag{10}
$$

**NFK for GANs**. As introduced in Section 3, we consider the EBM formulation of GANs. Given pre-trained GAN model, we use $D(\mathbf{x}; \boldsymbol{\theta})$ to denote the output of the discriminator $D$, and use $G(\mathbf{h})$ to denote the output of generator $G$ given latent code $\mathbf{h} \sim p(\mathbf{h})$. Then we have the energy-function defined as $E(\mathbf{x}; \boldsymbol{\theta}) = -D(\mathbf{x}; \boldsymbol{\theta})$. Based on the NFK formulation for EBMs, we can simply substitute

---

**Algorithm 6** `matrix_jacobian_product`

---

    **Input** Neural network model $f_{\boldsymbol{\theta}}$
    **Input** Input data $\mathbf{X} \in \mathbb{R}^{B \times D}$, where $B$ is batch size
    **Input** Input matrix $\mathbf{M}$
    **Input** Kernel type `kernel`, NFK or NTK
    **Output** $J_{\boldsymbol{\theta}}(\mathbf{X})\mathbf{M}$ for NTK, Fisher-vector-matrix-product $V_{\boldsymbol{\theta}}(\mathbf{X})\mathbf{M}$ for NFK
1: `mjp_fn = jax.vmap(jax.vjp)`
2: $\mathbf{P} =$ `mjp_fn`$(f_{\boldsymbol{\theta}}, \mathbf{X}, \mathbf{M})$
3: **if** `kernel` $=$ "NFK" **then**
4:     $\mathbf{P} = \text{diag}(\mathcal{I})^{-\frac{1}{2}}(\mathbf{P} - \mathbf{Z}_{\boldsymbol{\theta}}\mathbf{M})$
5: **end if**
6: Return $\mathbf{P}$

---

$E(\mathbf{x}; \boldsymbol{\theta}) = -D(\mathbf{x}; \boldsymbol{\theta})$ into Eq. 9 and Eq. 10 and derive the NFK formulation for GANs as below

$$
\begin{aligned}
U_{\mathbf{x}} &= \nabla_{\boldsymbol{\theta}} D(\mathbf{x}; \boldsymbol{\theta}) - \mathbb{E}_{\mathbf{h} \sim p(\mathbf{h})} \nabla_{\boldsymbol{\theta}} D(G(\mathbf{h}); \boldsymbol{\theta}) \\
\mathcal{I} &= \mathbb{E}_{\mathbf{h} \sim p(\mathbf{h})} \left[ U_{G(\mathbf{h})} U_{G(\mathbf{h})}^{\top} \right] \\
V_{\mathbf{x}} &= (\text{diag}(\mathcal{I})^{-\frac{1}{2}}) U_{\mathbf{x}} \\
\mathcal{K}_{\text{nfk}}(\mathbf{x}, \mathbf{z}) &= \langle V_{\mathbf{x}}, V_{\mathbf{z}} \rangle
\end{aligned}
\tag{11}
$$

Note that we use diagonal approximation of FIM throughout this work for the consideration of scalability. Also, since the generator of GANs is trained to match the distribution induced by the discriminator's EBM from the perspective of variational training for GANs, we could use the samples generated by the generator to approximate $\mathbf{x} \in p_{\boldsymbol{\theta}}(\mathbf{x})$, which is reflected in above formulation.

**NFK for VAEs, Flow-based Models, Auto-Regressive Models**. For models including VAEs, Flow-based Models, Auto-Regressive Models, where explicit or approximate density estimation is available, we can simply apply the classical Fisher kernel formulation as introduced in the main text.

**NFK for Supervised Learning Models**. In the supervised learning setting, we consider conditional probabilistic models $p_{\boldsymbol{\theta}}(\mathbf{y} \mid \mathbf{x}) = p(\mathbf{y} \mid \mathbf{x}; \boldsymbol{\theta})$. In particular, we focus on classification problems where the conditional probability is parameterized by a softmax function over the logits output $f(\mathbf{x}; \boldsymbol{\theta})$: $p_{\boldsymbol{\theta}}(\mathbf{y} \mid \mathbf{x}) = \exp(f^{\mathbf{y}}(\mathbf{x}; \boldsymbol{\theta})) / \sum_{\mathbf{y}} \exp(f^{\mathbf{y}}(\mathbf{x}; \boldsymbol{\theta}))$, where $\mathbf{y}$ is a discrete label and $f^{\mathbf{y}}(\mathbf{x}; \boldsymbol{\theta})$ denotes $\mathbf{y}$-th logit output. We then borrow the idea from JEM (Grathwohl et al., 2020) and write out a joint energy function term over $(\mathbf{x}, \mathbf{y})$ as $E(\mathbf{x}, \mathbf{y}; \boldsymbol{\theta}) = -f^{\mathbf{y}}(\mathbf{x}; \boldsymbol{\theta})$. It is easy to see that joint energy yields exactly the same conditional probability, at the same time leading to a free energy function:

$$
\begin{aligned}
E(\mathbf{x}; \boldsymbol{\theta}) &= -\log \sum_{\mathbf{y}} \exp(f^{\mathbf{y}}(\mathbf{x}; \boldsymbol{\theta})) \\
\nabla_{\boldsymbol{\theta}} E(\mathbf{x}; \boldsymbol{\theta}) &= -\sum_{\mathbf{y}} p_{\boldsymbol{\theta}}(\mathbf{y} \mid \mathbf{x}) \nabla_{\boldsymbol{\theta}} f^{\mathbf{y}}(\mathbf{x}; \boldsymbol{\theta})
\end{aligned}
\tag{12}
$$

Based on the NFK formulation for EBMs, we can simply substitute above results into Eq. 9 and Eq. 10 and derive the NFK formulation for GANs as below

$$
U_{\mathbf{x}} = \sum_{\mathbf{y}} p_{\boldsymbol{\theta}}(\mathbf{y} \mid \mathbf{x}) \nabla_{\boldsymbol{\theta}} f^{\mathbf{y}}(\mathbf{x}; \boldsymbol{\theta}) - \mathbb{E}_{\mathbf{x}' \sim p_{\boldsymbol{\theta}}(\mathbf{x}')} \sum_{\mathbf{y}} p_{\boldsymbol{\theta}}(\mathbf{y} \mid \mathbf{x}) \nabla_{\boldsymbol{\theta}} f^{\mathbf{y}}(\mathbf{x}'; \boldsymbol{\theta})
\tag{13}
$$

$$
\begin{aligned}
\mathcal{I} &= \mathbb{E}_{\mathbf{x} \sim p_{\boldsymbol{\theta}}(\mathbf{x})} \left[ U_{\mathbf{x}} U_{\mathbf{x}}^{\top} \right] \\
V_{\mathbf{x}} &= (\text{diag}(\mathcal{I})^{-\frac{1}{2}}) U_{\mathbf{x}} \\
\mathcal{K}_{\text{nfk}}(\mathbf{x}, \mathbf{z}) &= \langle V_{\mathbf{x}}, V_{\mathbf{z}} \rangle
\end{aligned}
\tag{14}
$$

## C.2 EFFICIENT LOW-RANK NFK/NTK APPROXIMATION VIA TRUNCATED SVD

We provide mode details on experimental observations on the low-rank structure of NFK and the low-rank kernel approximation algorithm here.

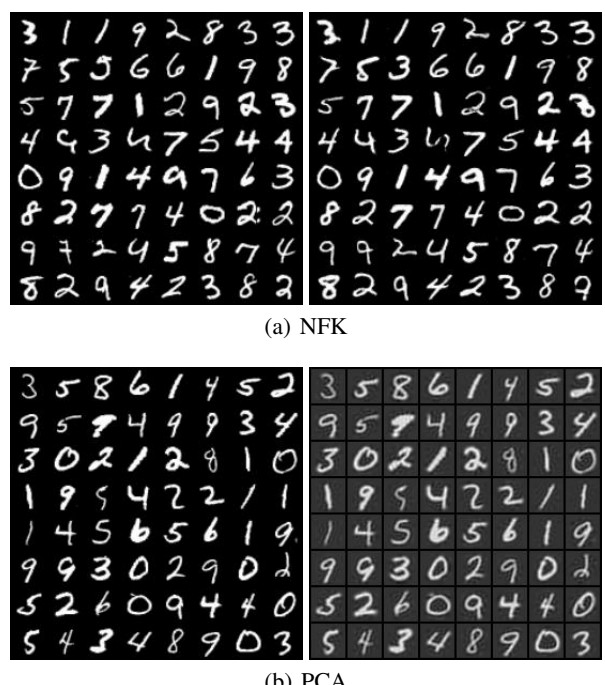

(a) NFK

(b) PCA

Figure 4: Inverting a DCGAN with 100d NFK embeddings (a), compared with image reconstruction with 100d PCA embeddings (b). In either case, the left plot corresponds to real test images and the right corresponds to the reconstructions. Note that NFK embeddings care capable of inverting a GAN by producing high quality semantic reconstructions. With PCA, embeddings with the same dimensionality produces more blurry reconstructions (thus less semantic).

**Low-Rank Structure of NFK**.

For supervised learning models, we trained a LeNet-5 (LeCun et al., 1998) CNN and a 3-layer MLP network by minimizing binary cross entropy loss, and then compute the eigen-decomposition of the NFK Gram matrix. For unsupervised learning models, we trained a small unconditional DC-GAN (Radford et al., 2016) model on MNIST dataset. We deliberately selected a small discriminator, which consists of 17K parameters. Because of the relatively low-dimensionality of $\theta$ in the discriminator, we were able to directly compute the Fisher Vectors for a random subset of the training dataset. We then performed standard SVD on the gathered Fisher Vector matrix, and examined the spectrum statistics. In particular, we plot the explained variance ration quantity, defined as $r_k = \frac{\sum_{i=1}^k \lambda_i^2}{\sum_{i=1}^n \lambda_i^2}$ where $\lambda_i$ is the $i$-th singular value. In addition, we have also visualized the top 5 principle components, by showing example images which have the largest projections on each component in Fig. 6.

Furthermore, we conducted a GAN inversion experiment. We start by sampling a set of latent variables from the generator's prior $\mathbf{h} \in p(\mathbf{h})$, and get a set of generated example $\{\mathbf{x}_i\}, \mathbf{x}_i = G(\mathbf{h_i}), i = 1, ..., n$. We then apply Algorithm 2 on the generated example $\{\mathbf{x}_i\}$ to obtain their NFK embeddings $\{\mathbf{e}(\mathbf{x}_i)\}$, and we set the dimension of both $h$ and $e$ to 100. We now have a compositional mapping that reads as $\mathbf{h} \to \mathbf{x} \to \mathbf{e}$. We then learn a linear mapping $\mathbf{W} \in R^{100 \times 100}$ from $\{\mathbf{e}(\mathbf{G}(\mathbf{h_i}))\}$ to $\{\mathbf{h}_i\}$ by minimizing $\sum_{i=1}^n \|\mathbf{h}_i - \mathbf{We}(\mathbf{G}(\mathbf{h_i}))\|^2$. In doing so, we have constructed an auto encoder from a regular GAN, with the compositional mapping of $\mathbf{x} \to \mathbf{e} \to \mathbf{h} \to \tilde{\mathbf{x}}$, where $\tilde{\mathbf{x}}$ is the reconstruction of an input $\mathbf{x}$. The reconstructions are shown in Figure 4 (a). Interestingly, the 100d SVD embedding gives rise to a qualitatively faithful reconstruction on real images. n contrast, a PCA embedding with the same dimension gives much more blurry reconstructions (eg., noise in the background), as shown in Figure 4 (b). This is a good indication that the 100d embedding captures most of the information about an input example.

**Power iteration of NFK as JVP/VJP evaluations**. Our proposed algorithm is based on the Power method Golub and Van der Vorst (2000); Bathe (1971) for finding the leading top eigenvectors of

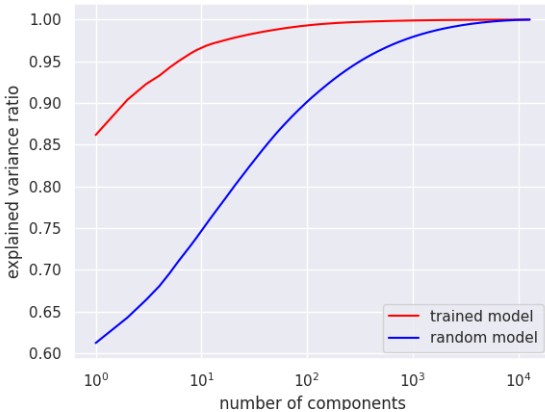

Figure 5: The low-rankness of the NFK on a DCGAN trained on MNIST. For a trained model, the first 100 principle components of the Fisher Vector matrix explains 99.5% of all variances. An untrained model with the same architecture on the other hand, demonstrates a much lower degree of low-rankness.

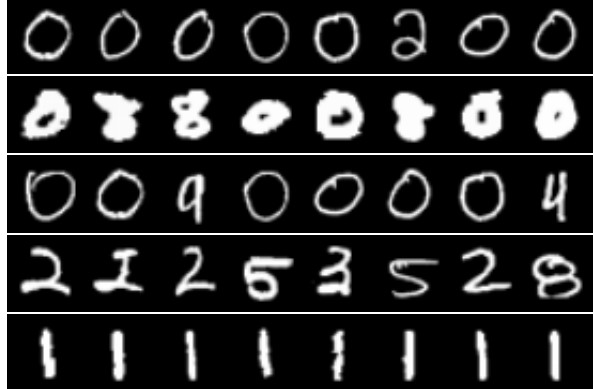

Figure 6: Images with the largest projections on the first five principle components. Each row corresponds to a principle component.

the real symmetric matrix. Starting from a random vector $\mathbf{v}_0$ drawn from a rotationally invariant distribution and normalize it to unit norm $\|\mathbf{v}_0\| = 1$, the power method iteratively constructs the sequence $\mathbf{v}_{t+1} = \frac{\mathbf{K}\mathbf{v}_t}{\|\mathbf{K}\mathbf{v}_t\|}$ up to $q$ power iterations. Given the special structure of $\mathbf{K}$ that it's a Gram matrix of the Jacobian matrix $J_{\boldsymbol{\theta}}(\mathbf{X}) \in \mathbb{R}^{D \times N}$, to evaluate $\mathbf{K}\mathbf{v}_t$ in each power iteration step we need to evaluate $J_{\boldsymbol{\theta}}(\mathbf{X})^\top J_{\boldsymbol{\theta}}(\mathbf{X})\mathbf{v}_t$, which can be decomposed as: (i) evaluating $\mathbf{z}_t = J_{\boldsymbol{\theta}}(\mathbf{X})\mathbf{v}_t$, and then (ii) $\mathbf{K}\mathbf{v}_t = J_{\boldsymbol{\theta}}(\mathbf{X})^\top \mathbf{z}_t$. Note that when $\mathbf{K}$ is in the form of NTK of neural networks, step (i) of evaluating $\mathbf{z}_t$ is a Vector-Jacobian-Product (VJP) and step (ii) is a Jacobian-Vector-Product (JVP). With the help of automatic-differentiation techniques, we can evaluate both JVP and VJP efficiently, which only requires the same order of computational costs of one backward-pass and forward-pass of neural networks respectively. In this way, we can reduce the Kernel matrix vector product operation in each power iteration step to one VJP evaluation and one JVP evaluation, without the need to computing and storing the Jacobian matrix and kernel matrix explicitly.

As introduced in Section. 3.2, we include detailed algorithm description here, from Algorithm. 3 to Algorithm. 6. In Algorithm. 3, we show the algorithm to compute the low-rank NFK embedding, which can be used as data representations. In Algorithm. 4, we present our proposed automatic-differentiation based truncated SVD algorithm for kernel approximation. Note that in Algorithm. 5 and 6, we only need to follow Equation. 3 to pre-compute the model distribution statistics, $\mathbf{Z}_{\boldsymbol{\theta}} = \mathbb{E}_{\mathbf{x}' \sim p_{\boldsymbol{\theta}}(\mathbf{x}')} \sum_{\mathbf{y}} p_{\boldsymbol{\theta}}(\mathbf{y}|\mathbf{x}) \nabla_{\boldsymbol{\theta}} f_{\boldsymbol{\theta}}^{\mathbf{y}}(\mathbf{x}')$, and FIM $\mathcal{I} = \mathbb{E}_{\mathbf{x}' \sim p_{\boldsymbol{\theta}}(\mathbf{x}')}[U_{\mathbf{x}'} U_{\mathbf{x}'}^\top]$. We adopt

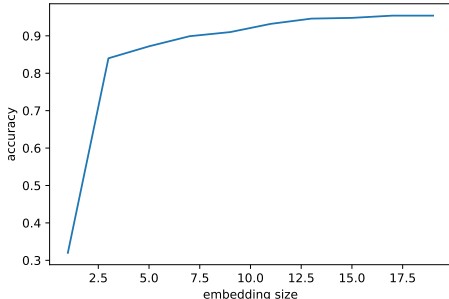

Figure 7: Linear probing accuracy on CIFAR10 with different number of principle components in embedding. We use our proposed low-rank approximation method to compute the embedding from the teacher model on CIFAR10 for knowledge distillation.

the EBM formulation of classifier $f_\theta(\mathbf{x})$ then replace the Jacobian matrix $J_\theta(\mathbf{X})$ with the Fisher vector matrix $V_\theta(\mathbf{X}) = \mathrm{diag}(\mathcal{I})^{-\frac{1}{2}}(J_\theta(\mathbf{X}) - \mathbf{Z}_\theta)$. Note that our proposed algorithm is also readily applicable to empirical NTK via replacing the FIM by the identity matrix.

## D  EXPERIMENTS SETUP

### D.1  QUALITY ANDEFFICIENCY OFLOW-RANKNFK APPROXIMATIONS

**Experiments on Computational Cost**. We randomly sample $N \in \{2^k : 7 \le k \le 16\}$ data examples from CIFAR-10 dataset, and compute top-32 eigenvectors of the NFK Gram matrix ($\mathbb{R}^{N \times N}$) by truncated SVD. We use same number of power iterations (10) in baseline method and our algorithm. We show in Fig. 3 the running time of SVD for both methods in terms of number of data examples $N$.

**Experiments on Approximation Accuracy**. We randomly sample 10000 examples and compute top-128 eigenvalues using both baseline methods and our proposed algorithm. Specifically, we compute the full Gram matrix and perform eigen-decomposition to obtain baseline results. For our implementation, we run 10 power iterations in randomized SVD.

### D.2  NEURAL FISHER KERNEL DISTILLATION

With the efficient low-rank approximation of NFK, one can immediately obtain a compact representation of the kernel. Namely, each example can be represented as a $k$ dimension vector. Essentially, we have achieved a form of kernel distillation, which is a useful technique on its own.

Furthermore, we can use $Q$ as an generalized form for teacher student styled knowledge distillation (KD), as in (Hinton et al., 2015). In standard KD, one obtain a teacher network (e.g., deep model) and use it to train a student network (e.g., a shallow model) with a distillation loss in the following format:

$$\mathcal{L}_{\mathrm{kd}}(\mathbf{x}, \mathbf{y}) = \alpha * \mathcal{L}_{cls}(f_s(\mathbf{x}), \mathbf{y}) + (1 - \alpha) * \mathcal{L}_t(f_s(\mathbf{x}), f_t(\mathbf{x})), \tag{15}$$

where $L_{cls}$ is a standard classification loss (e.g., cross entropy) and $L_t$ is a teacher loss which forces the student network's output $f_s$ to match that of the teacher $f_t$. We propose a straightforward extension of KD with NFK, where we modify the loss function to be:

$$\mathcal{L}_{\mathrm{nfkd}}(\mathbf{x}, \mathbf{y}) = \alpha * \mathcal{L}_{cls}(f_s(\mathbf{x}), \mathbf{y}) + (1 - \alpha) * \mathcal{L}_t(h_s(\mathbf{x}), Q_t(\mathbf{x})), \tag{16}$$

where $Q_t(\mathbf{x})$ denotes the $k$ dimensional embedding from the SVD of teacher NFK, for example $\mathbf{x}$. $h_s$ is a prediction head from the student, and $L_t$ is overloaded to denote a suitable loss (e.g., $\ell2$ distance or cosine distance). Equation 16 essentially uses the low dimension embedding of the teacher's NFK as supervision, inplace of the teacher's logits. There are arguable benefits of using $L_{nfkd}$ over $L_{kd}$.

For example, when the number of classes is small, the logit layer contains very little extra information (measured in number of bits) than the label alone, whereas $Q_t$ can still provide dense supervision to the student.

For the Neural Fisher Kernel Distillation (NFKD) experiments, we adopt the WideResNet-40x2 (Zagoruyko and Komodakis, 2016) neural network as the teacher model. We train another WideResnet with 16 layers as the student model, and keep the width unchanged. We run 10 power iterations to compute the SVD approximation of the NFK of the teacher model, to obtain the top-20 eigenvectors and eigenvalues. Then we train the student model with the additional NFKD distillation loss using mini-batch stochastic gradient descent, with 0.9 momentum, for 250 epochs. The initial learning rate begins at 0.1 and we decay the learning rate by 0.1 at 150-th epoch and decay again by 0.1 at 200-th epoch. We also show the linear probing accuracies on CIFAR10 by using different number of embedding dimensions in Figure. 7.

