# OpenReview forum: "Learning Representation from Neural Fisher Kernel with Low-rank Approximation"
_ICLR.cc/2022/Conference — ICLR 2022 Poster_

### Official Review · Reviewer_z7Tb · 2021-10-29

**Correctness:** 3
**Technical Novelty And Significance:** 3
**Empirical Novelty And Significance:** 2
**Recommendation:** 6
**Confidence:** 4

**Main Review:**

The paper proposes a formulation for the NFK for unsupervised (GANs and VAEs) and supervised neural networks. The kernels are defined as inner products of the Fisher vectors (derived from the Fisher score) computed from pre-trained neural networks.

Strength:

•	Nice idea to see extensions of kernels, here the Fisher kernel, to neural networks.

•	Similar formulations for supervised and unsupervised settings.

Weaknesses:

•	More motivation and derivations of the different Fisher scores $U_x$ (for GANs, VAEs, supervised) would be beneficial for understanding better the paper.

•	More discussions on the non-diagonal version of the Fisher information matrix models (see comments below) would be beneficial.

•	Discussion on the dependence of the quality of the NFK embedding on the quality of the pre-trained neural network.

•	Discussion on the choice of the low-dimensional embedding dimensionality $k$

General comments:

•	Using the diagonal of the Fisher information matrix (FIM) seems desirable from a computational reason, however a natural question is what happens if one tries to use the full matrix. Given the size of the parameters $\theta$ in a neural network, estimating the whole matrix would indeed be extremely computationally expensive, but by discarding them, one loses significant information. Could the authors comment on that? Is the diag of FIM related to other known concepts in statistics? Does using only the diagonal imply that the off-diagonal elements are zero meaning the parameters are orthogonal? How does this affect the results and interpretation?

•	Could the authors give the derivation of $U_x$ in eq (1) for GANs (I see part of the derivation is in the paper by Zhai et al, 2019)? For VAEs and supervised case, FIM $\mathcal{I}$ is the inner product using $U_x$, but for GANs it acts in the output space of the generator. Why is this the case (some explanations are given in the original paper but would be helpful to discuss this a bit more here)? The derivation of eq (4) would also be useful.

•	Low-rank structure of NFK and Alg 1: How does one choose the feature dimensionality $k$? Many methods that rely on kernels and manifold learning make the assumption of low-dimensionality/low-rankness and show that a small number of eigenfunctions is sufficient to reconstruct the input data. How is this different for NFK? The way I understand “low rank” is that the data has its own rank, which is low, and could potentially be learned. However here the authors input the dimensionality/rank $k$ which might be or not close to the true rank in real applications.

•	How does this work relate to the work of Belkin et al (2018) – “To understand deep learning we need to understand kernel learning”, where the authors look at other kernels (Laplacian and Gaussian)?

•	Could the approach be used for neural networks that are not pre-trained, as the neural tangent kernel NTK?

•	The experimental results are nice, however the focus on computation is not so relevant given that it only uses the diagonal of the Fisher information matrix. Comparisons using the whole matrix would also be needed. What error is used in Table 3 (MSE, MAE, RMSE)? The goal of the paper is to present a method for supervised and unsupervised settings, however in the results an example on semi-supervised is also presented. I wonder if the examples on semi-supervised and knowledge distillation could leave room to improve the supervised and unsupervised settings discussions, and potentially be moved to the Appendix?

Other comments:

•	Please update reference (Jaakkola et al): year, conference, also there should be no “et al” there are only two authors

•	Doesn’t, don’t, won’t, it’s , etc -> does not, do not, would not, it is

•	Both the concepts of “data representation” and “feature representation” are used. Do they always refer to the same thing? If yes, would be good to specify that.

•	Expression of $K_{fisher}$ => second $U$ should be subscript $z$ not $x$?

•	“FIM defined as the variance of the score …” -> the FIM matrix is defined between all pairs of parameters $\theta_i$ and $\theta_j$, so it should be a covariance?

•	Appendix Fig 3: Not sure I fully understand this example. Could one try the reconstruction of the digits using a simple method, such as PCA using the first 100 principal components as a baseline?

•	Not familiar with the “Fisher vector” terminology, except in image classification and the “Adversarial Fisher vector” from Zhai et al, 2019. Are there other references?


**Summary Of The Paper:**

The paper introduces a new kernel, the neural Fisher kernel (NFK), to learn data representations for generative models in both unsupervised and supervised settings. NFK extends the Fisher kernel to neural networks, and a low-rank approximation of NFK is used to scale to large datasets.

**Summary Of The Review:**

The paper presents interesting ideas of extending kernels, here the Fisher kernel, to neural networks. I find that the paper would need to make a stronger connection with existing literature (eg, Belkin et al, 2018) and give more motivation/discussion/intuition on the derivation of the different $U_x$s. My grading suggestion is a 6, as I believe the paper presents nice ideas but would require more work.

---

> ### Author Response · Authors · 2021-11-18
> **Response to reviewer z7Tb (part 5)**
>
> #### **Q11** For appendix Fig 3, could one try the reconstruction of the digits using a simple method, such as PCA using the first 100 principal components as a baseline?
> We have updated the manuscript with better explanation of this experiment, as well as results with PCA reconstructions.
>
> First of all, GAN inversion (reconstructing real examples) is typically done by training a deep encoder to map from generated samples to their latents, see [10] for an overview.
>
> In our experiment, we show that a linear encoder is sufficient for GAN inversion. We first construct a mapping as $\mathbf{h} \rightarrow \mathbf{x} \rightarrow \mathbf{e}$, where $\mathbf{h} ,\mathbf{x}, \mathbf{e}$ are GAN's latent variable, generated sample, and NFK embedding, respectively. We then learn a linear mapping $W \in R^{100\times d_h}$ for $\mathbf{e} \rightarrow \mathbf{h}$, on the set of generated samples. We can then use the compositional mapping of $\mathbf{x} \rightarrow \mathbf{e} \rightarrow \mathbf{h} \rightarrow \mathbf{\tilde{x}}$ to reconstruct a real example $\mathbf{x}$, ie, GAN inversion. This suggests 1) the 100d NFK embedding captures enough information about the inputs and 2) the NFK embedding can be linearly mapped to the semantic latent space of a GAN, both are non-tirival.
>
> In our updated results (Figure 4 (b)), we show that with PCA, a 100d projection gives much blurrier reconstructions, with clear noises in the background. This suggests that PCA embeddings are not able to correctly capture the data manifold, while NFK does not struggle with it.
>
> #### **Q12** Not familiar with the “Fisher vector” terminology, except in image classification and the “Adversarial Fisher vector” from Zhai et al, 2019. Are there other references?
> To the best knowledge of authors, the notion of Fisher vector comes from the work ([8]) and has been utilized in computer vision studies ([8,9,11,12]), that additional generative models (e.g. GMM) need to be learned, which is fundamentally different from our proposed approach.
>
> #### **Q13** Typos
> Thanks for pointing out the typos. We have updated the manuscript to correct the typos and references.
>
>
> #### References
>
> [1] Chen, Lin, and Sheng Xu. "Deep neural tangent kernel and laplace kernel have the same RKHS." arXiv preprint arXiv:2009.10683 (2020).
>
> [2] Geifman, Amnon, et al. "On the similarity between the laplace and neural tangent kernels." arXiv preprint arXiv:2007.01580 (2020).
>
> [3] Jacot, Arthur, Franck Gabriel, and Clément Hongler. "Neural tangent kernel: Convergence and generalization in neural networks." arXiv preprint arXiv:1806.07572 (2018).
>
> [4] Ghorbani, Behrooz, et al. "When do neural networks outperform kernel methods?." arXiv preprint arXiv:2006.13409 (2020).
>
> [5] Lee, Jaehoon, et al. "Finite versus infinite neural networks: an empirical study." arXiv preprint arXiv:2007.15801 (2020).
>
> [6] Li, Zhiyuan, et al. "Enhanced convolutional neural tangent kernels." arXiv preprint arXiv:1911.00809 (2019).
>
> [7] Duchi, J., Hazan, E., & Singer, Y. (2011). Adaptive Subgradient Methods for Online Learning and Stochastic Optimization. Journal of Machine Learning Research, 12, 2121–2159.
>
> [8] Sánchez, Jorge, et al. "Image classification with the fisher vector: Theory and practice." International journal of computer vision 105.3 (2013): 222-245.
>
> [9] Perronnin, Florent, et al. "Large-scale image retrieval with compressed fisher vectors." 2010 IEEE computer society conference on computer vision and pattern recognition. IEEE, 2010.
>
> [10] Xia, Weihao et al. "Gan inversion: A survey".
>
> [11] Azim, Tayyaba, and Sarah Ahmed. Composing Fisher Kernels from Deep Neural Models: A Practitioner's Approach. Springer, 2018.
>
> [12] Yoo, Donggeun, et al. "Fisher kernel for deep neural activations." arXiv preprint arXiv:1412.1628 (2014).

---

> ### Author Response · Authors · 2021-11-18
> **Response to reviewer z7Tb (part 4)**
>
> #### **Q8** The experimental results are nice, however the focus on computation is not so relevant given that it only uses the diagonal of the Fisher information matrix.
> While the diagonal approximation could alleviate the computational challenge raised by FIM, it can not resolve the core technical issues in this work. Note that the Fisher kernel is defined as $\mathcal{K}(\mathbf{x}, \mathbf{z}) = \nabla_{\theta}\log p_{\theta}(\mathbf{x})^\top \mathcal{I}^{-1}\nabla_{\theta}\log p_{\theta}(\mathbf{z})$. Even without the presence of FIM $\mathcal{I}$, where NFK would degenerate to the empirical NTK formulation, $\mathcal{K}(\mathbf{x}, \mathbf{z})=\langle \nabla_{\theta} f(\mathbf{x};\theta), \nabla_{\theta} f(\mathbf{z};\theta) \rangle$ where $f(\mathbf{x};\theta)=\log p_{\theta}(\mathbf{x})$, the high-dimensionality nature of the feature vector $V_{\mathbf{x}}=\nabla_{\theta}f(\mathbf{x};\theta) \in \mathbb{R}^{|\theta|}$  still poses a significant challenge in terms of computational efficiency and scalability. We briefly illustrate the technical difficulties case by case here.
> * To exploit the kernel formulation in the primal form, i.e. training a linear model on top of the feature vector associated with the kernel, as considered in Zhai et al, 2019, where explicit Jacobian matrix $\mathbf{V} \in \mathbb{R}^{N \times |\theta|}$ needs to be explicitly instantiated for a batch of $N$ data examples. For classification setup with $C$ classes, the time and space complexity would be at least $O(N|\theta| + C|\theta|)$. This approach is infeasible whenever we have a large neural network, which is quite common in modern machine learning practice. For example, the Wide-ResNet-40x2 model used in our work has about $50M$ parameters, which results in a Jacobian matrix with shape $(50000, 5\times 10^7)$ on CIFAR-10, and it would even be hard to make a small batch of the feature matrix to fit into the GPU memory.
> * To exploit the kernel formulation in the dual form, i.e. leveraging the Gram matrix $\mathbf{K} \in \mathbb{R}^{N \times N}$, which scales quadratically with respect to the number of data examples, would lead to the inherent scalability issues in kernel methods. Please also see our response to reviewer 2 (9i1k)'s Q2 for technical difficulties in this case.
> * Our proposed efficient low-rank kernel approximation algorithm along with the experimental evaluation of efficiency mainly focus on addressing the scalability issue raised by the above computational difficulties, rather than handling the challenges of approximating FIM. We do believe that a better approximation of FIM would lead to better performance of NFK, and we would consider continuing to push in this direction in future works.
>
> #### **Q9** What error is used in Table 3 (MSE, MAE, RMSE)?
> The reported numbers in Table 3 are the classification accuracies of the student network on CIFAR-10. We've updated the manuscript to improve the clarity in the caption text of Table 3.
>
> #### **Q10** The goal of the paper is to present a method for supervised and unsupervised settings, however in the results an example on semi-supervised is also presented. I wonder if the examples on semi-supervised and knowledge distillation could leave room to improve the supervised and unsupervised settings discussions, and potentially be moved to the Appendix?
> The goal of this work is to propose a principled approach to representation extraction from pre-trained neural network models, which can be either pre-trained supervisedly or unsupervisedly. Therefore we focus on evaluating the quality of the extracted feature representation in terms of (i) the effectiveness for downstream tasks and (ii) the sample efficiency, by following common protocols in representation learning works. The semi-supervised learning setup is mainly for validating the sample efficiency of the extracted representation from NFK. For supervised learning, we considered using knowledge distillation as the experimental setup to evaluate the informativeness of the learned representation from NFK, by using it as the distillation target to transfer the knowledge from the teacher network to the student network.

---

> ### Author Response · Authors · 2021-11-18
> **Response to reviewer z7Tb (part 3)**
>
> #### **Q6** How does this work relate to the work of Belkin et al (2018)?
> Thanks for pointing out the reference! We've updated our manuscript to provide more detailed discussions on the connections with the Laplace kernel, Gaussian Kernel, and NTK in the related works section and appendix. We also briefly summarize the connections and discrepancies here.
> * The focus of Belkin et al (2018) and this work are different. Belkin et al (2018) focus on investigating the unique generalization properties of over-parametrized neural networks via contrastive studies on kernel methods, while our work focus on proposing a novel representation extraction framework from pre-trained neural networks through a novel NFK formulation to unify both supervised learning models and unsupervised learning models.
> * On the connections between the Laplace kernel and the analytical NTK. Belkin et al (2018) empirically showed that the Laplace kernel and neural networks had similar performance in fitting random labels and the learning dynamics of neural networks more closly resembles the Laplace kernel than the Gaussian kernel. Further theoretical works[1,2] showed that if we assume that when data are normalized to the hypersphere $\mathbf{x} \in \mathbb{S}^{d-1}$, the analytical NTK (under standard NTK parametrization) for fully connected neural network with ReLU activations and the Laplace kernel share the same reproducing kernel Hilbert space (RKHS). On the other hand, the RKHS associated with the Gaussian kernel only includes infinitely smooth functions.
> * On the connections with the NFK. Though the Laplace kernel is closely related to the analytical NTK, it only holds true based on the assumptions that the kernels are restricted to the sphere and the NTK is under the standard parametrization ([3]), i.e. under the expectation over randomly initialized weight parameters. On the contrary, the setup of NFK in this work significantly differs from the above assumption that NFK focus on **trained** neural networks, while the Laplace kernel, the Gaussian kernel are essentially **fixed** kernels. As studied in recent works[4,5,6], feature learning matters a lot in the comparative evaluation between NTK and neural network models, as a significant generalization gap exists especially for complex datasets and neural network architectures.
>
>
> #### **Q7** Could the approach be used for neural networks that are not pre-trained, as the neural tangent kernel NTK?
> Yes, however, the quality of extracted representation from NFK depends on how well the model is trained to model the data distribution. As shown in Figure 1. right, we compared the low-rankness of the NFK of a trained neural network against a randomly initialized model, and the trained neural network has a much lower effective rank.

---

> ### Author Response · Authors · 2021-11-18
> **Response to reviewer z7Tb (part 2)**
>
> #### **Q4** How is the low-rank assumption of manifold learning different for NFK?
> Thanks for this interesting question! The essential idea behind our proposed approach shares the same assumption of the manifold hypothesis as manifold learning methods, however, this work focused on approaching the problem from a different angle. Different from other manifold learning methods or generative models (e.g. PCA, Kernel PCA, AutoEncoders, etc), which aim to learn a low-dimensional representation from *data*, the goal of NFK along with proposed low-rank embedding is to serve as a principled framework to extract a low-dimensional representation from the *pre-trained/learned model* , which can be either unsupervised learning models or supervised learning models. That being said, the Riemannian manifold defined by the class of probabilistic models, where the local metric (FIM $\mathcal{I}$) is given by the pre-trained/learned weights $\theta$, can be seen as a proxy for the true unknown data manifold, through the mapping from data space $\mathbf{x} \in \mathcal{X}$ to $V_{\mathbf{x}} \in \mathbb{R}^{|\theta|}$. As discussed in Section 3.2, *good* models (with effective inductive bias, pre-trained well, etc) should align well with true data manifold, thus we have good reasons to believe that the corresponding NFK should reflect the nature of data manifold and exhibit a low-rank structure. We believe that the NFK framework could provide new perspectives on uncovering the hidden knowledge in the pre-trained neural network models.
>
> #### **Q5** How does one choose the feature dimensionality $k$? Here the authors input the dimensionality/rank $k$, which might be or not close to the true rank in real applications.
> We agree that the selected $k$ in this work's experiments might not coincidence with the true rank of real data, however, the goal of this work is a principled approach to extracting effective feature representation from pre-trained models, which suggests that we could choose to set $k$ as large as possible while remaining in the computational budget, depending on the specific real application. We showed in our experiments, e.g. Table 1, that by using $k = 128$ we could recover the full Fisher vector's representational effectiveness, which can be viewed as an estimation of the upper-bound of the true rank of data. The choice of $k$ should be better evaluated through the effectiveness of the low-rank NFK embedding $\mathbf{e}_{\text{nfk}}  \in \mathbb{R}^k$, in terms of representational power and sample efficiency in the down-stream applications of interest, given the unknown nature of true intrinsic dimensionality of real-world data.

---

> ### Author Response · Authors · 2021-11-18
> **Response to reviewer z7Tb (part 1)**
>
> Thank you for your positive feedback and careful reviews! We appreciate your constructive comments and we are glad to hear that you found our work interesting.
>
> #### **Q1** Could the authors comment on the diagonal approximation of FIM?
> As pointed out by the reviewer, we adopted the diagonal approximation of FIM in this work for the sake of scalability. However, taking quote from Section 4 in (Jaakkola and Haussler, 1998), "The role of the information matrix is less significant; indeed, in the context of logistic regression models, the matrix appearing in the middle of the feature vectors relates to the covariance matrix of a Gaussian prior, as show above. Thus, asymptotically, the information matrix is immaterial."
> The diagonal approximation of the FIM assumes the parameters to be independent, though it is not true, it works quite well in practice. This is also the standard approach in the Fisher Vector literature, see [8]. Moreover, the Adagrad optimizer introduced diagonal approximation to the Hessian and it is one of the most effective optimization methods in practice, while noting the close connection between the Hessian and the FIM that FIM is the expected Hessian of $\log p(\mathbf{x};\theta)$ over $\mathbf{x} \sim p(\mathbf{x};\theta)$. We would also like to consider other feasible approximations to the FIM, e.g. the block-diagonal K-FAC approximation, in the future works.
>
> #### **Q2** Could the authors give the derivation of $U_{\mathbf{x}}$ in eq (1) for GANs and the derivation of eq. (4)?
> We have updated our manuscript to include an extra section in the appendix (Section C.1) to present detailed derivation of the NFK formulations for EBMs, GANs, VAEs and supervised learning models. The derivation of eq. 1 and eq. 4 basically follow the derivation of NFK formulation of EBMs, and we showed that only the energy function formulations for different models are different and we just need to substitute $E(\mathbf{x})$ to obtain the final results.
>
> #### **Q3** Why is $\mathbf{x} \sim G(\mathbf{h})$ used in NFK for GANs, $\mathbf{x} \sim p_{\theta}(\mathbf{x})$ is used in VAEs and supervised learning models?
> These formulations are essentially the same with different practical approximations. By the definition of FIM $\mathcal{I}=\mathbb{E}\_{\mathbf{x} \sim p_{\theta}(\mathbf{x})} \left[ U_{\mathbf{x}}U_{\mathbf{x}}^{\top} \right]$, we should use $\mathbf{x} \sim p_{\theta}(\mathbf{x})$ to evaluate FIM, where $p_{\theta}(\mathbf{x})$ is the model distribution. We have updated our manuscript to make it clearer for the notations.
> * For GANs, from the view of the EBM formulation of GAN, the generator with maximum entropy regularization is trained to match the distribution induced by the EBM $E(\mathbf{x})=-D(\mathbf{x})$, where $D(\mathbf{x})$ is the output of discriminator. Thus we can use the generating distribution of the generator, denoted as $G(\mathbf{h}), \mathbf{h} \sim p(\mathbf{h})$ in paper, to approximate the model distribution $p_{\theta}(\mathbf{x}) \propto \exp(-E(\mathbf{x}))$. We have also updated our paper to include more introductin on this point in the newly included section in the appendix.
> * For VAEs, we use $p_{\theta}(\mathbf{x})$ to denote the data generating distribution of the VAE model, namely the distribution induced by $\mathbf{x} = \text{decoder}(\mathbf{h}), \mathbf{h} \sim p(\mathbf{h})$.
> * For supervised learning models, $p_{\theta}(\mathbf{x})$ denotes $p_{\theta}(\mathbf{x}) \propto \exp(-E(\mathbf{x}))$ where $E(x) = -\operatorname{logsumexp}(f(\mathbf{x}))$, thus to obtain samples $\mathbf{x} \sim p_{\theta}(\mathbf{x})$ we need to resort to MCMC methods to sample from the EBM. For practical considerations, we use empirical data distribution to approximate $p_{\theta}(\mathbf{x})$ in this work, as pointed out in Section 3.1.2 in ths paper.

---

### Official Review · Reviewer_9i1k · 2021-11-02

**Correctness:** 4
**Technical Novelty And Significance:** 3
**Empirical Novelty And Significance:** 3
**Recommendation:** 6
**Confidence:** 3

**Main Review:**

strengths:
The general idea of using the linear space associated with the kernel feature map for extracting data representation is interesting. Simplifying the representation through low-rank approximation of the kernel matrix is an interesting technique and may have a good impact on representation learning. Using the power method for computing the low-rank approximation through automatic differentiation could have applications that go beyond the proposed approach.

weaknesses:
The authors should specify better what is new and what has been already done. For example, it looks like using Fisher vectors in representation learning and approximating kernels through truncated SVD are not new ideas. Also, the authors could have spent some more words to outline the main differences and advantages of using the Fisher kernel instead of the standard Tangent kernel. It is not clear if the main contribution of the paper is the extension to unsupervised learning or the proposed low-rank approximation.

questions:
- Is the approximation of the fisher tangent kernel the key novelty of the paper? If so, what is the technical challenge compared with approximating other kernels?
- I like the idea of using automatically computed Jacobians in the power method. Is this a new idea? Can this be applied to the standard tangent kernel?
- The experiments do not compare the proposed approach with non-fisher similar methods. Would it be possible to run a quantitative comparison with the tangent kernel (at least for the supervised case)?
- Are there computational gains associated with using the Fisher kernel for its low-rank approximations to train a model or is the proposed method only suitable for obtaining more intuitive representations?
- Is Montecarlo Sampling needed in the proposed method? And what are the computational cost and feasibility limits of that power iteration approach?
- In the third line of the last paragraph of p2, Ux I Ux should be Ux I Uz, right?


**Summary Of The Paper:**

The paper proposes a new strategy for extracting compact and intuitive representations of the data from a neural network. The approach exploits the feature map associated with the kernel representation of a neural network. In particular, the authors consider the Neural Fisher
Kernel and propose a series of linear approximations to make their approach scalable.


**Summary Of The Review:**

An interesting idea but the authors should specify better what is new and what comes from existing work.

---

> ### Author Response · Authors · 2021-11-18
> **Response to reviewer 9i1k (part 3)**
>
> #### **Q9** In the third line of the last paragraph of p2, Ux I Ux should be Ux I Uz, right?
> Right, thanks a lot for pointing out the typo! We have updated the manuscript to correct the typos.
>
> #### References
>
> [1] Goodfellow, Ian. "Efficient per-example gradient computations." arXiv preprint arXiv:1510.01799 (2015).
>
> [2] Mu, Fangzhou, Yingyu Liang, and Yin Li. "Gradients as features for deep representation learning." arXiv preprint arXiv:2004.05529 (2020).
>
> [3] Arora, Sanjeev, et al. "On exact computation with an infinitely wide neural net." (2019)
>
> [4] Samarin, Maxim, Volker Roth, and David Belius. "On the empirical neural tangent kernel of standard finite-width convolutional neural network architectures." (2020)

---

> ### Author Response · Authors · 2021-11-18
> **Response to reviewer 9i1k (part 2)**
>
> #### **Q5** Would it be possible to run a quantitative comparison with the tangent kernel?
> We compared against NTK on both theoretical side and empirical side in this work. One major difficulty in quantitative comparison with NTK is that NTK is not well defined for unsupervised learning models, and thus we will use the tangent features as the analogy to NTK in unsupervised learning models. We provide more detailed analysis on the comparison with NTK as below:
> * Unsupervised learning. As discussed in the paper, there's no well-established theory for NTK in unsupervised learning settings yet, while our proposed NFK formulation is immediately applicable to both unsupervised learning and supervised learning settings. Therefore there's no proper comparison between NFK and NTK in this case. As for the baseline methods([2]) which utilize tangent space features (i.e. gradients) as data representation, we compared the performance of NFK with [2] in Table. 1 quantitatively.
> * Supervised learning. We compare NFK to NTK in terms of both empirical performance and efficiency here.
>   * Performance: We compare our proposed NFK to both analytical NTK and empirical NTK. For analytical NTK in infinite-width neural networks, there exists a non-negligible performance gap between kernel regression using analytical NTK (under infinite-width regime) and real-world neural networks (finite-width), as studied in previous literature (Table. 1, [3]), NTK (CNTK) of the largest model (21-layer CNN) can only achieve $64.09\%$ accuracy compared to its original $75.57\%$ accuracy on CIFAR-10. For empirical NTK for finite-width neural networks, as reported in previous works (Table 1, [4]) there also exists a large generalization gap between linearized model using NTK and original model, e.g., the linearized ResNet-18 model attains $56.7\%$ accuracy compared to the original $91.0\%$ accuracy, which exhibits a $34.3\%$ generalization gap. On the other hand, our proposed low-rank NFK embedding is able to fully recover the original supervised trained CNN's (WideResNet-40x2) performance by using up to $20$ principal dimensions (see Figure.6 in Section. D.1 in Appendix), which shows that our proposed low-rank NFK embedding approach could effectively close the performance gap and capture the information in original neural networks.
>   * Efficiency: The shape of the NTK tensor is $N^2C^2$, where $N$ is the number of data examples and $C$ is the number of classes, it scales quadratically with the number of classes $C$ in supervised learning models, which makes it hard to scale to large scale real-world machine learning tasks. The size of our proposed NFK is $N^2$, which is invariant to the number of classes.
>
>
> #### **Q6** Are there computational gains associated with using the Fisher kernel for its low-rank approximations to train a model or is the proposed method only suitable for obtaining more intuitive representations?
> We would like to first distinguish the notion of the model for a clearer discussion context here. For the model which is given as a pre-trained model, e.g. the GANs and Wide-ResNet considered in the experiments in this work, NFK and its low-rank approximation does not affect the training phase of neural network models, instead, it is used to extract representation from given off-the-shelf pre-trained models. For the model which is used in the linear probing evaluation, which is essentially a one-layer linear model on top of the extracted representation, we have huge computational gains coming from the low-rank approximation compared to the full NFK formulation, a reduction from $V_{\mathbf{x}} \in \mathbb{R}^{|\theta|}$ to $\mathbf{e}_{\mathbf{x}} \in \mathbb{R}^k$ where $k \ll |\theta|$.
>
> #### **Q7** Is Montecarlo Sampling needed in the proposed method?
> Yes, Monte Carlo sampling is needed in several places where empirical data distribution is used to approximate the true data distribution to estimate the FIM $\mathcal{I}$ as well as the eigenfunctions and eigenvalues of NFK, as discussed in the paper in Section 3.
>
> #### **Q8** What are the computational cost and feasibility limits of that power iteration approach?
> The computational cost of each power iteration would be the same as the cost of one forward pass and one backward pass in the vanilla neural network training process. The feasibility limit would depend on the complexity of the pre-trained model and the data distribution, which could affect the rate of convergence of estimating the eigenfunctions and eigenvalues of NFK, and the bottleneck of space complexity in each power iteration would be mostly determined by $\max(N, K)$ where $N$ is the number of data examples in one sampled mini-batch and $K$ is the desired dimensionality of low-rank kernel approximation.

---

> ### Author Response · Authors · 2021-11-18
> **Response to reviewer 9i1k (part 1)**
>
> Thank you for your positive feedback and thoughtful suggestions. We are glad to hear that you found our work interesting and potentially impactful.
>
> #### Manuscript update
>
> To address the concern on the clarity of the novelty, we included a general summary of this work's novel contributions at the end of the introduction section, as well as more discussions on the connection to existing works, in our revised manuscript.
>
> #### **Q1** Is the approximation of the NFK the key novelty of the paper?
> The proposed efficient NFK low-rank approximation algorithm is indeed *one of* the key novel contributions made in this work. As highlighted at the end of Section 1 in our revised manuscript, we made both theoretical and empirical contributions. Theoretically, we introduced a novel kernel formulation, NFK, as a unified framework for neural networks models in both supervised learning and unsupervised learning settings. Empirically, we proposed a novel low-rank kernel approximation algorithm, which allows us to obtain a compact feature representation in a highly efficient and scalable way.
>
>
> #### **Q2** What is the technical challenge compared with approximating other kernels?
> In contrast to other kernels, we briefly discuss the major technical challenges of NFK low-rank kernel approximation from two perspectives here.
> * Inherent scalability issue in kernel approximation: As mentioned in Section 3.2, no matter which kernel formulation to choose, existing off-the-shelf kernel approximation algorithms require explicit instantiation of either the Gram matrix $\mathbf{K} \in \mathbb{R}^{N \times N}$ or the feature design matrix $\mathbf{V} \in \mathbb{R}^{N \times D}$, where $N$ is the number of data exmples and $D$ is the dimensionality of the feature vector associated with the kernel. Considering modern machine learning setups where $N$ can be very large, it imposes a non-trivial scalability issue in kernel approximation problem.
> * Technical difficulties specific to NFK: The feature vector associated with NFK, namely the Fisher vector $V_{\mathbf{x}} \in \mathbb{R}^{|\mathbf{\theta}|}$, has the same dimensionality as the number of parameters of the neural network model $|\theta|$, which can be tremendously large given the scale of modern neural networks. It naturally raised the practical challenge that we need to scale the kernel approximation to the scenario where both $N$ and $D$ ($D=|\theta|$ for NFK) can be large. On the other hand, different from some classical layer-activations based kernels (like NNGP kernel), the computation of Fisher vector $V_{\mathbf{x}}$ involves per-example gradient evaluation, which also imposes a practical challenge in implementation ([1]).
> * Compared to (empirical) Neural Tangent Kernel (NTK), for which the above arguments also hold true, a significant discrepancy is that NFK involves the Fisher Information Matrix (FIM) $\mathcal{I} \in \mathbb{R}^{|\theta| \times |\theta|}$ normalization. Note that NFK was proposed as a unified and principled kernel framework due to the limitations of NTK, and as discussed in Section 3, handling $\mathcal{I}$ in NFK should not be seen as unnecessary extra burden. On the contrary, it provides a natural normalization scheme over the gradients space and better numerical stability.
>
> #### **Q3** I like the idea of using automatically computed Jacobians in the power method. Is this a new idea?
> To the best knowledge of authors, using the JVP/VJP trick to compute a low-rank kernel approximation, as well as the low-rank embedding, is novel. We also updated the paper to include more discussions of related works including the automatic-differentiation methods used in second-order optimization algorithms.
>
> #### **Q4** Can the proposed kernel approximation algorithm be applied to the standard tangent kernel?
> Yes, the proposed low-rank kernel approximation algorithm is readily applicable to the NTK, by just replacing the FIM $\mathcal{I}$ by the identity matrix. We've updated our manuscript to make this clearer.

---

### Official Review · Reviewer_945A · 2021-11-05

**Correctness:** 3
**Technical Novelty And Significance:** 3
**Empirical Novelty And Significance:** 3
**Recommendation:** 6
**Confidence:** 4

**Details Of Ethics Concerns:**

No Ethics Concerns.

**Main Review:**

This paper is interesting and technically sound, its novelty comes from two perspectives. First, it considers to represent/modify neural networks from the view of kernels. The proposed NFK serves as a unified kernel for both supervised and unsupervised learning models. Besides, the authors take advantage of the low-rank structure of NFK to avoid directly handling unmanageably high dimensional vectors.

1. The effectiveness of the proposed NFK is mainly evaluated based on CIFAR-10 dataset, how is the performance of the proposed method on other datasets?

2. The authors mention some representative unsupervised learning tasks such as denoising. Have the authors tried to combine the proposed NFK with the simple CNN for image denoising/super-resolution/enhancement?

3. Some figures could be used to help to better explain the framework of the proposed method and also its effectiveness.

4. Some terms such as VAEs and GANs should be defined,  for readers without too much background knowledge may be confused.

**Summary Of The Paper:**

The paper investigates the representation of modern neural networks from the perspective of kernels by extracting features from pre-trained network models. The authors show the effectiveness of the proposed neural fisher kernel (NFK) for both unsupervised and supervised learning tasks. A low-rank approximation strategy of NFK is adopted to reduce computational burdens.

**Summary Of The Review:**

The paper is technically interesting, but the authors should carefully revise their experimental section to address the concerns.

---

> ### Author Response · Authors · 2021-11-18
> **Thanks for your review**
>
> Thank you for your positive feedback and valuable suggestions! We are glad to hear that you found our work interesting and technically sound.
>
> #### **Q1** How is the performance of the proposed method on other datasets?
> 1. Our design of experiments and choice of datasets largely follow existing works([1,2]), and we have shown consistent results on both the CIFAR-10 dataset (Table 1, 3) and SVHN dataset (Table 2).
> 2. We also conduct experiments on a diverse set of tasks,  including unsupervised learning, semi-supervised learning and knowledge distillation, to support the motivation of NFK as an *unified* kernel formulation for neural network models in different settings.
> 3. But we agree that experimental results on more datasets, especially large scale ones, will be interesting. Due to the computational cost reasons, we consider leaving it as future work.
> <!-- other datasets beyond existing evaluations on CIFAR-10 and SVHN can be benificial, we're working on more experimental results on extra datasets to continue the improvement on this work. -->
>
> #### **Q2** Have the authors tried to combine the proposed NFK with the simple CNN for image denoising/super-resolution/enhancement?
> Thanks for pointing out the potential applications on conditional image generation tasks. We haven't tried them, but we agree that it's an interesting direction to explore in future works.
>
> We mentioned denoising autoencoders in the paper appendix to introduce the related works on existing representation learning methods, along with other self-supervised learning models. The representation extracted from NFK could be combined with other methods to be used in the downstream tasks of interest.
>
> #### **Q3** Some figures could be used to help to better explain the framework.
> Thanks for the suggestion and we have updated the manuscript to include a figure to illustrate the overall algorithmic pipeline of this work.
>
> #### **Q4** Some terms such as VAEs and GANs should be defined, for readers without too much background knowledge may be confused.
> Thanks for pointing it out. We have updated our manuscript to introduce the technical preliminaries of generative models used in this work.
>
> #### References
> [1] Tim Salimans, Ian Goodfellow, Wojciech Zaremba, Vicki Cheung, Alec Radford, and Xi Chen. Improved techniques for training gans. arXiv preprint arXiv:1606.03498, 2016.
>
> [2] Zhai, Shuangfei, et al. "Adversarial fisher vectors for unsupervised representation learning." arXiv preprint arXiv:1910.13101 (2019).

---

> > ### Comment · Reviewer_945A · 2021-11-29
> > **Thanks for your response**
> >
> > Dear authors,
> >
> > Thanks for your responses to my concerns. I am satisfied with your answers.

---

### Author Response · Authors · 2021-11-18
**Revision Summary**

We thank all reviewers for the positive assessment of our work and helpful suggestions for the improvements. We have taken these reviews seriously and have updated our manuscript to address the concerns, as summarized here.
* Overview figure. We have included an overview figure (Figure. 1) to demonstrate the general framework.
* Clarity of novel contributions. We have expanded our introduction section to include a brief summary (Section. 1) of novel contributions made in this work.
* More technical background introduction. We have covered more technical introduction on related works on generative models and kernel methods (Section 2 & Appendix) to make the connections and notations clearer.
* Detailed derivations of NFK formulations. We have included an extra section in the appendix (Section. C.1) to present the detailed derivations of the NFK formulations for GANs, supervised learning models, etc.
* More comparative experimental results. We have included additional experimental results (Figure. 4) to compare the image reconstruction quality with the PCA baseline.
* We have made the edits to correct the typos and update the references pointed out by reviewers.

---

### Author Response · Authors · 2021-11-29
**Thanks for all your reviews**

Dear AC and all reviewers,

Thanks again for your positive feedback and constructive suggestions,  which have helped us improve the quality and clarity of our paper!  Since it is close to the end of the discussion window,  please feel free to let us know if there are any additional concerns that we could address. We would greatly appreciate your time and feedback!

---

### Decision · Program_Chairs · 2022-01-20

**Decision:**

Accept (Poster)

**Comment:**

This is a borderline paper which elicited much discussion.
The paper proposes to extract features from pre-trained networks through Kernel functions. It develops the idea of Fisher kernels for Neural networks calling it NFK. The methodology applies to both supervised and un-supervised setting.
The paper shows that proposed kernel has low rank structure and serves as the basis for developing an algorithm for computing the kernel on large datasets. The idea of extending Fisher kernels, their efficient computation, and investigating their usage in both Supervised and Unsupervised are some of the key strengths of the paper.
The reviewers though appreciative suggested (1) several new experiments, (2) inclusion of more background work related to Power method
and (3)  have more technical discussion clarifying the contributions related to background.
The author(s) during rebuttal tried to incorporate most of the suggestions in the revised draft.

Since there was consensus on the novelty, the detailed discussions, and the results of the additional experiments, one could potentially accept this paper if there is space. The results will be interesting will be those who investigate the interplay of kernel methods and Deep Networks.